



# Blade surface pressure and drag measurement of a blade section on a 4.3 MW turbine with trailing edge flaps

Helge Aagaard Madsen[1], Alejandro Gomez Gonzalez[2], Thanasis Barlas[1], Anders Smærup Olsen[1], Sigurd Brabæk Ildvedsen[1], and Andreas Fischer[1]

[1]Department of Wind and Energy Systems, Technical University of Denmark, 4000 Roskilde, Denmark
[2]Technology Development, Siemens Gamesa Renewable Energy, 7330 Brande, Denmark.

**Correspondence:** Helge Aagaard Madsen (hama@dtu.dk)

**Abstract.** In this paper we present the measurements of local aerodynamic sectional characteristics on a full-scale rotor blade with a novel add-on instrumentation comprising a wake rake, a pressure belt, and a five hole Pitot tube. The general objective of the research work is to provide information on the differences between airfoil performance in wind tunnel flow and on a full-scale rotor. Although pressure belt measurements have been performed in earlier studies, this is the first campaign to use

a wake rake at the full-scale level. We present the wake rake development and testing in the wind tunnel and on a rotating test rig which finally led up to installation on the 4.3 MW turbine. A more specific objective with the campaign has been to characterize the aerodynamic performance of a trailing edge active flap system installed on one of the blades. The short measurement campaign of two days comprised measurements of the flaps actuated at constant time intervals of 60 s between fully retracted and activated with a control set point of 75% of full deflection. The lift and drag characteristics are compared

with a similar flap actuation in a wind tunnel experiment. Both the relative change in lift and drag coefficients as a function of flap actuation correlate well with wind tunnel measurements, but the absolute drag levels measured on the rotor are higher than the wind tunnel data. During the measurement campaign, it was also demonstrated that it is possible to clearly measure the increased drag from adding a roughness tape at the leading edge of the airfoil. This illustrates the potential use of the measurement system to capture the effect of variations of the local aerodynamic performance at full-scale even for elements at

boundary layer scale, e.g. the impact of roughness or the positioning of add-ons such as vortex generators.

## 1   Introduction

The lack of detailed measurements to characterize the aerodynamics and aeroelasticity of full-scale turbines is a barrier to further reliable turbine development and upscaling (Veers et al., 2023). In particular, inflow and blade surface pressure measurements on turbines operating in turbulent atmospheric inflow are crucial data for a deeper understanding and insight into

how the airfoil characteristics (lift, drag, and moment coefficients) measured in wind tunnels can be transferred and used for the design of full-scale wind turbine blades. This is due to the quite different operating conditions for an airfoil section tested in the steady, 2D wind tunnel flow and the unsteady, turbulent 3D flow experienced on the wind turbine blade.



A common approach for the determination of aerodynamic coefficients for blade design and load calculations is to use a weighted average of measured or simulated aerodynamic characteristics for free-transition and tripped (forced-transition) wind
tunnel airfoil data, where the weighting ratio is based on experience from previous blade designs and full-scale rotor tests. Reasons for the need of such empirical approaches can be found in the lack of knowledge of, e.g. the characteristics of the airfoils operating at the high Reynolds numbers found in normal field operation, the roughness state of the surface of the blades (e.g. clean vs. dirty blades), and outer-flow induced bypass transition at varying levels of free-flow turbulence. Together, these three mechanisms have an important influence on the boundary layer transition characteristics. Detailed measurements of the
transition position based on readings of surface microphones mounted flush on a full-scale rotor have shown that the level of atmospheric turbulence moves the transition toward the leading edge (Madsen et al., 2019).

Hopefully, measured airfoil characteristics on blade sections of a full-scale wind turbine blade in real operating conditions can lead to less empiricism in the use of wind tunnel airfoil data in industrial blade design.

Blade surface pressure measurements on full-scale turbines have so far mainly been performed employing sensors and
measurement equipment built into the blade and on specific test turbines (Madsen et al., 2010; Medina et al., 2011). This has hindered such measurements from being carried out on industrial-scale turbines to a greater extent and limiting the access to the detailed aerodynamic data mentioned above. However, with the fast development in sensor technology, e.g. the considerable reduction of the size of the sensors, it is now possible to design compact and autonomous add-on measurement systems, which do not require any integration of measurement equipment during the manufacturing of the blade.

The need for autonomous add-on measurement systems has led to activities at different research institutes to develop such measurement systems. The pressure belt instrumentation developed at DTU was presented by Madsen et al. (2022), while the development and test of an MEMS-based system was described by Barber et al. (2022). The measurement system of Fredebohm et al. (2023) comprises a system with pressure sensors embedded in thin glass fiber shells, which are glued to the rotor blades.

Another recent comprehensive full-scale experiment was conducted by TNO and LM Wind Power within the TIADE project
Boorsma et al. (2024), Fritz et al. (2024). Blade surface pressure measurements were planned at four spanwise locations on a 3.8 MW rotor. At the inboard section, with radius 15 m with access inside from the blade, a traditional pressure measurement installation was used. Further outboard with limited space inside the blade, a fiber optic pressure system was installed. However, it was found that blade surface strain complicated accurate pressure surface measurements.

The main objective of the present article is to present a further development and application of the system originally used
in a measurement campaign carried out in 2021 (Madsen et al., 2022), including the measurement of the surface pressure of a blade section with one belt and the inflow with a five-hole Pitot offset radially from the belt by approximately 1 m.

However, measuring the blade surface pressure distribution does not provide full information for the derivation of the airfoil characteristics. In addition to the pressure drag derived from the pressure distribution, there is the viscous drag from the shear stresses in the boundary layer. This is commonly measured with a so-called wake rake measuring the momentum loss in the
near wake behind the trailing edge of blade section.

In the present article, we present the further development of the measurement system used in the 2021 campaign (Madsen et al., 2022), comprising the design and test of a blade-mounted wake rake. In addition, the pressure belt was redesigned





from a 16 to a 32-tap pressure belt while the inflow was measured with a five-hole Pitot tube as in the previous campaign. With this instrumentation, the sectional airfoil characteristics can be characterized to a high degree of detail, and thus also the aerodynamic performance of an active trailing edge flap system installed on the turbine for this campaign.

In Section 2 the wake-rake development work is discussed along with an overview of the other components of the measurement system. The measurement campaign carried out on the SG-4.3-120 DD turbine in July 2022 is briefly described in Section 3 followed by a presentation of the results in Section 4. Then a discussion and outlook are provided in Section 5 and conclusions are drawn in Section 6.

## 2 Experimental set up

### 2.1 The pressure belt

The DTU pressure belt and inflow measurement system developed in the last 5-6 years includes the manufacturing of different prototypes that have been tested in various wind tunnel campaigns. The system was first used in full-scale testing campaigns on a 4.3 MW turbine in 2021 (Madsen et al., 2022) and now in the present experiment a modified pressure belt design with 32 taps is used.

However, it can be mentioned that more recently in April 2024 a completely new pressure belt design manufactured in an extrusion process was used in the downwind experiment on the GE 1.5 MW turbine operated in a downwind configuration in a test campaign in April 2024. The main objective of applying the system in this project was to provide experimental data characterizing the complex aerodynamics of the blade tower interaction of a downwind rotor (Bortolotti, 2024), (Bortolotti et al., 2025).

### 2.2 The wake rake

The development and testing of the wake rake comprised three major activities:

* Prototype design and testing parameter variations in the Poul la Cour wind tunnel (PLCT) at DTU
* Mid scale testing on a rotating test rig
* Full scale testing on the SWT-DD-120 4.3 MW turbine

#### 2.2.1 Prototype design and wind tunnel tunnel testing

Wind tunnel tests were conducted to estimate the main dimensions of the wake rake design.

– Distance between the trailing edge (TE) and the wake rake
– Spacing of individual probe tubes
– Spanwise location of the wake rake
– Wake rake angle relative to flow





– Width of the wake rake

The test section in the PLCT is 2 m in span and 3 m width. For this campaign, a blade section with a 1 m chord and a 2 m span was used; see Fig. 1. The blade section was equipped with a trailing edge morphing flap, as one of the crucial design parameters to determine in the PLCT tests is the width of the wake rake so it can capture the deficit with the flap in its maximum and minimum deflection positions. The instrumentation comprised standard pressure orifices in the midspan and two five-hole Pitot tubes at 0.5-m and 1.5-m span positions measured from the floor of the tunnel.

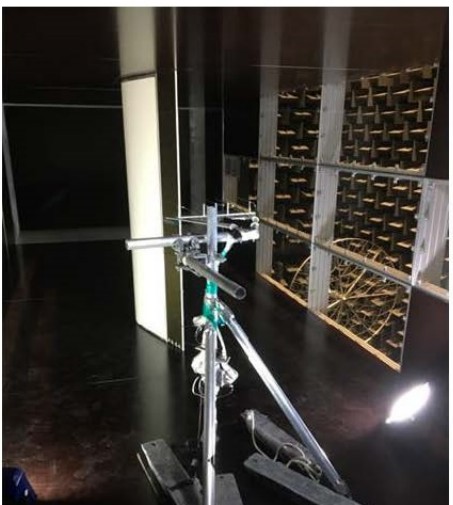
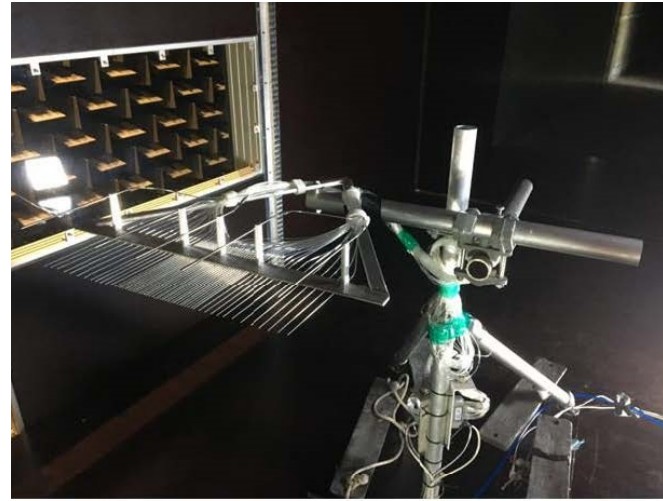

**Figure 1.** Photos of the set-up of the test wake rake in the Poul la Cour wind tunnel at DTU. The wake rake was installed behind a blade section with a span of 2 m, a 1 m chord and with a morphing trailing edge flap.

The test parameters in the wind tunnel were chosen to match conditions similar to the wake rake test on the rotating test rig. The test parameters chosen are as follows:

– $U_0$ = 20 m/s (Re = $1.3 \times 10^6$) and 30 m/s (Re = $1.9 \times 10^6$)
– Distance behind TE (from trailing edge to end of total pressure tubes): 0.1 m, 0.3 m, 0.5 m and 1.0 m
– Vertical position: 1.0 m (behind pressure orifices), 1.15 m (slightly above pressure orifices), 1.38 m (maximum height, closer to the connecting tubes for the five-hole Pitot tube)
– Wake rake angle: 0 deg (perpendicular to the flow), 8 deg (following the profile)
– AoA : 8 deg, 15 deg
– Flap deflection: 0 mm, -20 mm, + 20 mm

The drag coefficient $c_d$ is found using the method of Jones (1936) and may be expressed as:





$$c_d = 2 \int\limits_{y_{min}}^{y_{max}} \sqrt{c_{p,tot} - c_{p,stat}} \left[ 1 - \sqrt{c_{p,tot}} \right] dy, \tag{1}$$

with

$$c_{p,tot} = \frac{H_0 - p_0}{0.5\rho U_0^2}, c_{p,stat} = \frac{p - p_0}{0.5\rho U_0^2}, \tag{2}$$

where $y_{\max} - y_{\min}$ is the length of the integration path of the wake rake velocity deficit, $p_0$ the static pressure, $p$ the dynamic pressure, $H_0$ the total head, $\rho$ the density, and $U_0$ the velocity of the free stream.

From the wind tunnel test results, the following conclusions and parameter settings were derived:

- The $c_d$ measurement is only slightly affected by the downstream distance from the trailing edge (0.1 m to 0.5 m)
- The $c_d$ measurement is only slightly affected by rotating the rake with the profile, at least not for the value of angle of attack of 8 deg (It is also a matter of making the end of the tube conical)
- The $c_d$ is only slightly affected by the pressure orifices, but largely affected by the five-hole Pitot tube connection
- The wake width is approximately 0.3 m at $\alpha = 8$ deg for a flap deflection of $\delta = \pm 20$ mm.

Based on the findings and conclusions of the wind tunnel tests, the following dimensions of the wake rake were chosen:

- position of the head of Pitot tubes 0.3 m after the trailing edge
- wake rake width 0.4 m (above the measured 0.3 m to ensure covering impact of unsteadiness in the wake)
- four static tubes in line with the total pressure tubes
- 56 total pressure tubes (5 mm and 10 mm spacing, respectively)

A photo of the manufactured wake rake is shown in Fig. 2. Inside the wake rake plastic tubes are connected to the pressure probes and led out at both ends of the rake to a pressure scanner outside the tunnel.

### 2.2.2 Mid scale testing on a rotating test rig

During the development of flap technology at DTU from 2010 to today, it was recognized that the increase in complexity connected to the transfer of technology from wind tunnel testing to the full-scale flap prototype test level was too large and associated with high risks. Therefore, in 2015, it was decided to build a so-called rotating test rig (RTR) at the DTU to test flaps in the real operational environment but on a smaller scale (Barlas et al., 2014). The RTR is built on a 100 kW turbine platform where the rotor is replaced by a 10 m pitch-able long boom carrying a 2 m span blade section at the tip, Fig. 3. In fact, a blade section for tests in the PLC tunnel can be tested directly on the RTR.

The wake rake was mounted on the blade section with a frame of circular carbon tubes attached to the aluminum frame inside the blade section; see the photo on the left in Fig. 3. The 2 m long rectangular blade section + end caps has an active



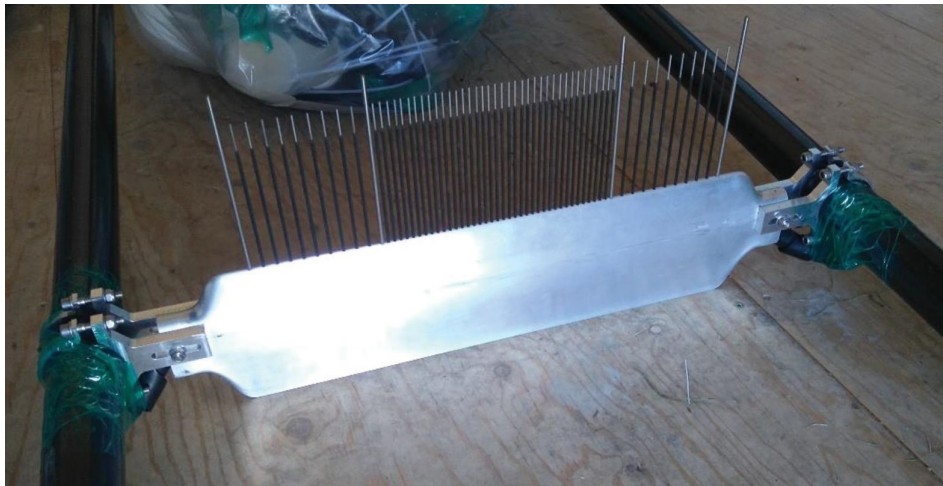

**Figure 2.** Photo of the final manufactured wake rake.

trailing edge flap system with serrations. However, the serrations were covered with tape as seen in Fig. 3 as it is expected that the flow will not be uniform in the position of the wake rake due to the serrations and thus make the drag measurement uncertain. Pressure taps were installed mid-span on the blade section in a standard procedure from inside, as the blade section was manufactured in two parts. Inflow was measured with two five-hole Pitot tubes, left photo in Fig. 3 and the wake rake probes were connected to a differential pressure scanner positioned inside the blade section.

Detailed tests were carried out in different wind conditions, different pitch settings of the blade section, and with various flap activations. In addition, boundary layer manipulators were tested to measure the impact on the drag. An example is the test of vortex generators in combination with surface roughness, as shown in Fig. 4. The lift and drag coefficients were derived from the pressure and drag measurements in combination with the measured inflow angle and relative velocity from the five-hole Pitot tubes. The impact of the roughness is clearly measured and is about 0.005 in most of the operating region outside stall conditions. For confidential reasons, the absolute values in Fig. 4 are left out.

### 2.2.3 Wake rake attachment on the full scale blade

Based on the experience of mounting and operating the wake rake on the blade section for the RTR and considering the quite different attachment conditions on the full-scale blade as there is no access to the interior of the blade, it was decided to design and manufacture a new frame for attachment of the wake rake on the full-scale blade; see CAD designs in Fig. 5a. CFD flow simulations were used to predict and adjust the wake rake position to optimally cover the width of the wake deficit, as shown schematically in Fig. 5b.



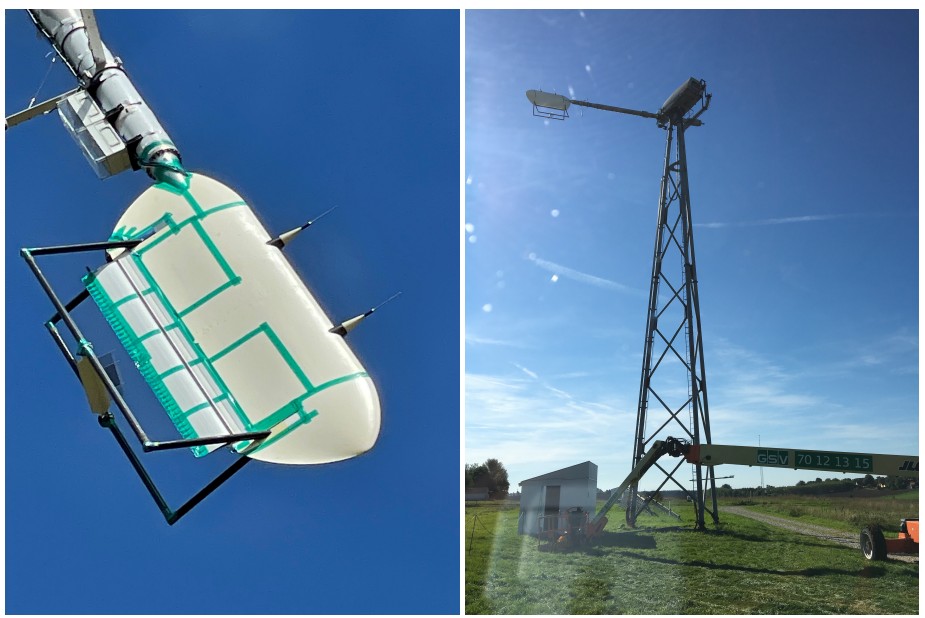

**Figure 3.** Left: The wake rake attached to the 2 m span blade section with carbon fiber circular tubes fastened inside the blade section to the aluminum frame. Right: The blade section on the 9m long rotating boom during the RTR tests.

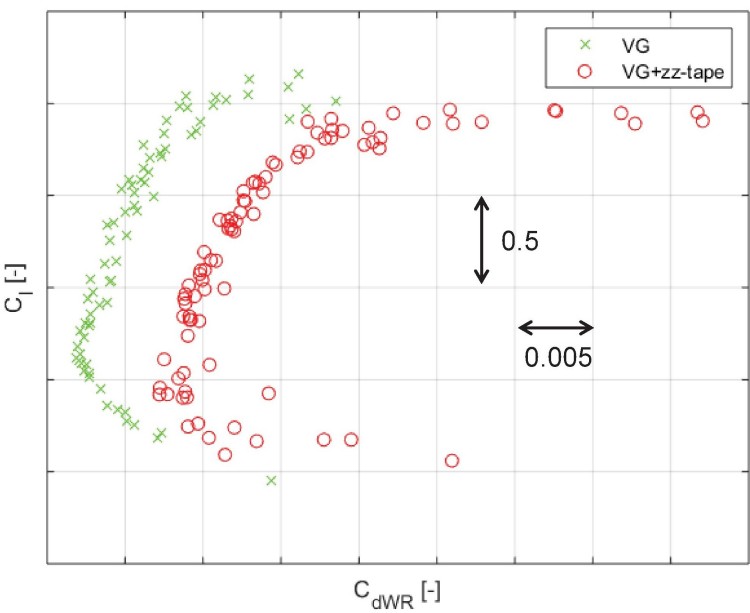

**Figure 4.** The measured impact of added surface roughness with zig zag tape in combination with vg´s on a 21 % thick airfoil section with an active flap system. The RTR was operated with a constant rpm at different wind speeds and the lift and drag coefficients were derived based on the measured blade pressure distribution, the wake rake measurements and the inflow measured with the five-hole Pitot tubes.




As preparatory work for the mounting of the wake rake, some interface pads were glued onto the blade one week before the actual mounting of the wake rake support structure. These interface pads were designed to carry shear and tension loads

from the support structure without the need to drill holes in critical structural areas of the blade. A single hole was drilled in the blade at a location that was not relevant to the structural integrity of the blade, where a tension bolt was inserted to stiffen the entire wake rake structure to ensure stability during operation. Circular carbon rods were still used as the main elements of the frame as seen in Fig. 5. The four 16-channel mini differential pressure scanners connected to the wake rake tubes were mounted (taped) on the support structure at each end of the wake rake. The photos in Figs. 6a and 6b show the wake rake after

installation.

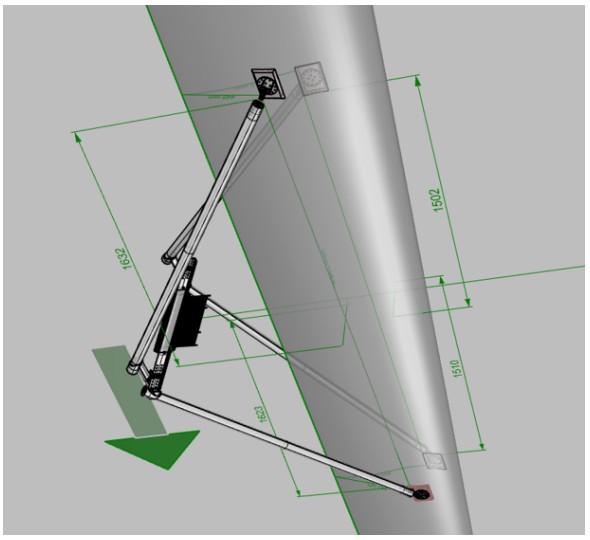

(a) Wake rake and blade mounting approach.

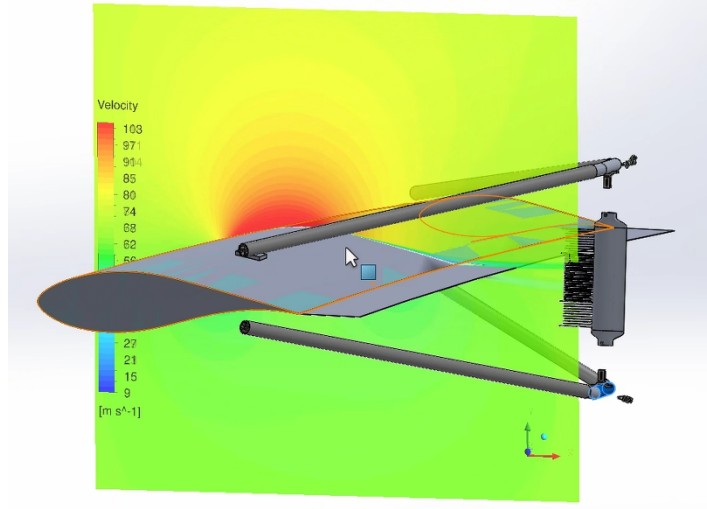

(b) CFD simulations for alignment of with expected velocity deficit.

**Figure 5.** CAD design of wake-rake attachment on the full scale blade.

## 2.3    Set-up of full scale testing

### 2.3.1    Installation of equipment

The installation of the wake rake system on the turbine was carried out on the morning of 28 July 2022. The complete installation was performed in approximately 4-5 hours from the basket of a truck-mounted lift. A weather window of relatively low

wind speeds and no rain was chosen, with the primary focus of allowing good working conditions for the installation of the equipment, as well as to avoid rain during the duration of the tests due to the weather sensitivity of the Pitot and the wake rake probes.

The wake rake was structurally secured to the interface pads with screws and the entire system was tensioned with an arrangement of tension ropes to ensure high stiffness during the operation of the turbine. The pressure belt and Pitot inflow sensor



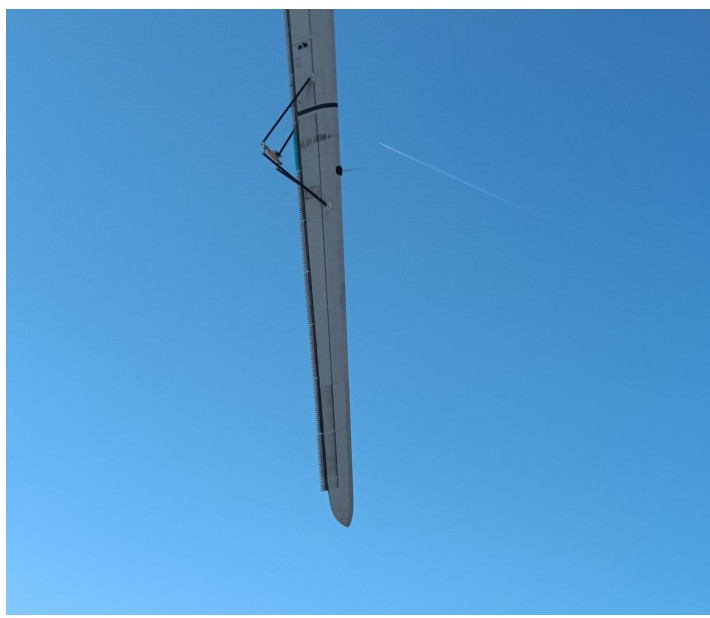

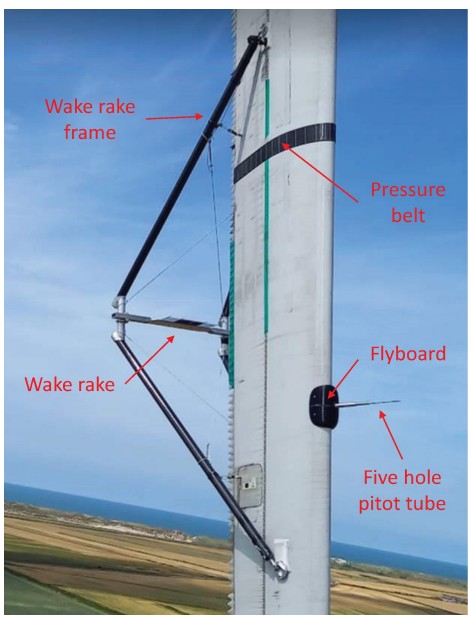

(a) Overview of the full setup on the blade 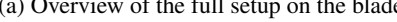 (b) Close-up view of the wake-rake setup.

**Figure 6.** The instrumented blade section with the wake rake, the pressure belt and the flyboard with the data acquisition system and carrying the five hole Pitot tube.

were subsequently mounted and connected to the data acquisition system. The wake rake was centered around a spanwise location approximately 50 m from the root of the blade, with the pressure belt and Pitot probe mounted 800 mm offset towards the inboard and outboard area, respectively.

Finally, GoPro cameras were installed on interface locations close to the wake rake setup which had been prepared in advance to have video overview of the full test setup while performing the test, see the video link in the *"Video supplement"* at the end of the article.

### 2.3.2 Turbine and flap system

The SG-4.3-120 DD is a prototype wind turbine, fully instrumented for load and power performance validation campaigns, located in Høvsøre, Denmark (see site layout in Fig. 7). The specific wake rake measurements were carried out during phase 4 as part of the VIAs project (Gomez Gonzalez, 2019) , for which one of the blades had previously been equipped with an active flap system (AFS). The main objective of the VIAs project was to redesign, manufacture, and validate an AFS for the rotor blades of wind turbines and to develop the methods and instrumentation necessary for experimental and numerical validation.

In what follows a short description of the AFS is given, but the reader is referred to (Gomez Gonzalez et al., 2022, 2023) for further details. The outer portion of the blade (span location 38 m – 58 m) is retrofitted with the active flap (internal reference FT008rev10) with remote pneumatic actuation pressure. Note that the AFS includes trailing edge serrations, which is important



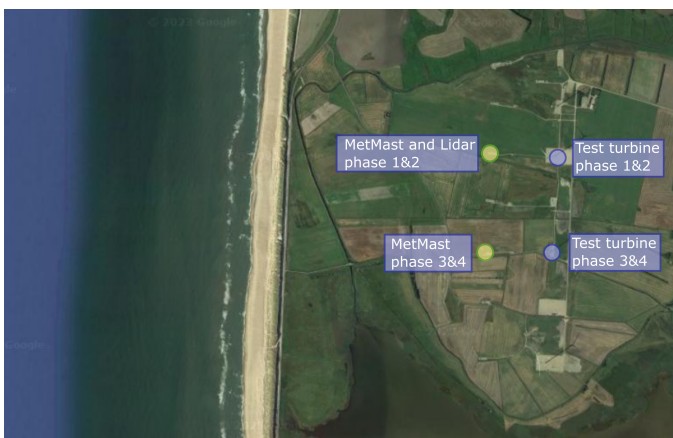

**Figure 7.** Test site layout. Map data taken from GoogleMaps copyright - 2025

to highlight as will be discussed below. The pressure supply system is located in the hub, and air is supplied to the flap via hoses inside the blade that extend from the hub and into the interior of the blade and up to a distance from the root where physical access is no longer possible due to the internal dimensions of the blade. At this location a smaller diameter hose is continued on the outside of the blade flush with the trailing edge until it reaches the inlet of the active flap located 38 m from the root.

The pressure supply system is closed loop regulated and for given pressure set points (specified between 0% and 100% for the lower and upper limit of flap activation, respectively). The active flap can be activated in swap mode at predefined time intervals and set points, or in a closed-loop manner based on the azimuthal position of the blades.

     For the measurement with the wake-rake the trailing edge serrations were covered with tape extending in the spanwise direction, rendering the trailing edge of the AFS effectively straight. This modification, which is visible in Fig. 6b as a green

tape in the serration area, extends symmetrically in and outboard of the wake-rake position. The reason for this modification is the following: when performing wind tunnel measurements of airfoils with serrations, it is known that the integrated momentum deficit of the wake, i.e. the drag, has a periodic behavior with a spanwise wave length equal to the spacing between teeth of the serrations. This behavior is due to the local upwash and downwash induced by the vortical system created behind the serrations through their interaction with the bounday layer flow. When performing a wind tunnel measurement one must therefore traverse

the wake-rake along the span of the airfoil to ensure covering the drag variations along multiple teeth. If no traverse is done, it is not possible to know if the wake-rake position is in the "high" or the "low" area of the momentum deficit. Because wake traversing is not possible in the full-scale test, the serrations are covered with tape to avoid the spanwise drag variations. However, one consequence of this is that the drag coefficient measured in the full-scale test will be slightly larger than that measured in the wind tunnel due to the additional area coverage of the AFS. In the future, it may be considered to benchmark

the results against wind tunnel tests where the same modification is performed on the serrations.





### 2.3.3 Turbine instrumentation

The turbine is equipped with various sensors, all logging continuously at 25 Hz. The main sensors of interest for this particular test are the pressure set point of the AFS, the rotor speed and power of the turbine, as well as the pitch and azimuth angle of the blade. Atmospheric conditions such as wind speed at different heights, temperature, and atmospheric pressure are collected 205 using a meteorological mast (met-mast) and a Lidar located at the position marked with *phase 1 & 2* in Fig. 7, as the met-mast used specifically for phase 4 of the test was no longer in operation the days when the wake-rake test was conducted. During the whole campaign, the wind speed is also recorded with the turbine's own sonic nacelle anemometer.

### 2.3.4 Pressure belt

In the present experiment, an update of the pressure belt design with 32 channels was used compared to the 16 channels of 210 the pressure belt used in the 2021 experiment (Madsen et al., 2022). However, the basic belt design is the same as in the 2021 experiment with a printed plastic fixture in which the plastic tubes are pressed and fixed. For further details on pressure belt installation and pressure scanners, the reader is referred to Madsen et al. (2022).

### 2.3.5 Flyboard and the data acquisition system

As for the pressure belt, the flyboard with the data acquisition system, attached to the leading edge of the blade and carrying 215 the five-hole Pitot tube, is also the same as in the 2021 experiment and the reader is again referred to Madsen et al. (2022) for further details. The pressure scanners on the wake rake are in a similar fashion connected in the data acquisition system on the flyboard.

The scanning rate of the pressure belt, the wake rake, and five hole Pitot tube plus one three-dimensional accelerometer is 100 Hz and the data are stored on the acquisition board and in parallel sent to a laptop on the ground by a WiFi system.

### 2.3.6 Data synchronization and processing

Both the wind turbine data (sampled at 25 Hz) and the aerodynamic measurement system data (sampled at 100 Hz) were post-processed and synchronized. The synchronization process has to be done with care as the different signals of the turbine and the flyboard/wakerake setup are collected with independent data acquisition systems. The first step of synchronization is based on the GPS time stamp of all measurements. The second step of synchronization was based on alignment of the acceleration 225 signals of the blade-mounted data acquisition system with the hub-mounted measurement of the azimuthal position of the blade. For this second step, a wind turbine start-up sequence is used, where the acceleration level of the blade in the radial direction (i.e. the centrifugal acceleration) is overlayed with the azimuthal position, and the time correlation of these signals is optimized. Because the start-up sequence is a slow procedure, this type of synchronization is very well suited for wind turbine experiments, where the azimuthal variations of the measured variables are of interest.

The data of all signals collected by the blade-mounted data acquisition, including the pressure belt, the wake rake, the inflow Pitot probe, and the acceleration signals were first processed by performing a down-sampling in the form of a block averaging



of every 4 consecutive samples, effectively reducing the sample rate to 25 Hz (the same acquisition rate as for the wind turbine signals). Subsequently, a moving average with a window length of 10 samples (i.e. 0.4 s) was applied to all data sets. An example of pressure data collected by the belt and the wake rake before and after these data processing steps is shown in Fig.

8. For the sake of clarity, these plots show only every second pressure port of both the belt and the wake rake, for an exemplary time trace of 60 s of duration. The characteristic sinusoidal variation of all signals is due to the rotation of the turbine with a 1P frequency in combination with shear in inflow and minor instantaneous yaw errors.

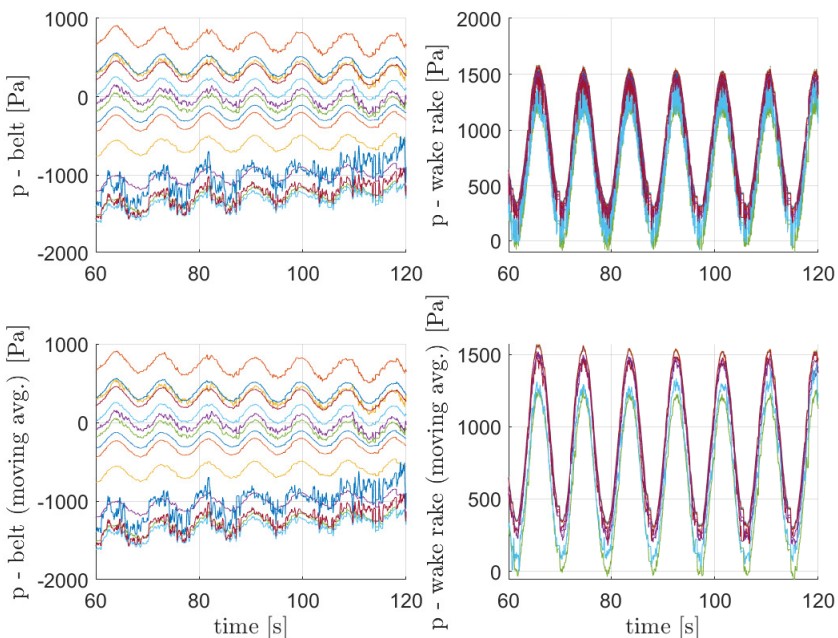

**Figure 8.** Exemplary time traces of different pressure signals from the pressure belt and total pressure wake rake probes with and without block and moving average

For the evaluation of the lift coefficient, the first step is to integrate the individual pressure measurements of the belt along the contour of the airfoil. Except for the viscous forces, this integral yields a total aerodynamic force on the airofil section. The

total force is initially projected into normal and parallel directions to the chord using the information of the airfoil geometry. These two components are then transformed to obtain the components perpendicular and parallel to the flow using the angle of attack measurement of the Pitot probe, representing the lift and pressure drag of the airfoil, respectively. The pressure drag is not further processed because the total drag is obtained directly from the velocity deficit measured with the wake rake.

To obtain the lift coefficient, the lift force is normalized with the local chord in the position of the pressure belt and with

the dynamic pressure $Q$. The dynamic pressure would normally be measured from the Pitot probe. However, the Pitot dynamic pressure measurement was faulty during the campaign, with values clearly underestimating expected levels by up to 40%. Therefore, the dynamic pressure $Q$ was obtained from the difference between the outermost total and static pressure probes of





the wake rake, i.e. the probes that are least influenced by the prescence of the airofil and which are located in the "flat" region of the velocity deficit.

The angle of attack of the airfoil is measured with the Pitot probe using the correlation coefficients for the 5-hole measurements obtained in a wind tunnel calibration. Furthermore, the measurement is corrected for the mounting angle of the probe ($8\,\mathrm{deg}$ with respect to the chord line). Finally, the local impact of the airfoil up-wash on the measurement location of the probe (which extends 480 mm into the flow) is estimated using CFD simulations of the airfoil with and without active flap actuation. Usually, for an airfoil without an active flap, a single linear correlation is found between the free wind inflow angle (i.e. the

angle of attack) and the angle measured directly at the probe. This correlation is mostly linear within the linear range of the lift coefficient. However, for an airfoil with an active flap, this correlation varies depending on the actuation level of the flap due to its impact on the overall bound circulation level of the airfoil. The approach taken by the authors was to perform one CFD correlation for the maximum active flap activation position and one CFD correlation for the position where the flap was inactive. For all intermediate positions, linear interpolation was used, using the flap activation level as the interpolation factor.

As will be discussed later, the uncertainty of the angle of attack measurements is one of the most important factors in the full analysis, as this value depends on the accuracy of the calibration and measurement, the potential geometry deviations during mounting, and the upwash correction for the operational turbine.

### 2.3.7   Uncertainty analysis

Although a comprehensive uncertainty quantification study is beyond the scope of this work, an attempt to list important

sources of uncertainty is presented here.

Some components of the measurement and processing chain are estimated to have a minimal contribution to the results. These include the pressure scanner measurement accuracy (which is 0.1 % full scale for the 100 millibar measurement range). The Pitot tube and wake rake total/static pressure tubes have also been calibrated in the wind tunnel with insignificant mounting uncertainty, so the accuracy depends only on the pressure scanners. Based on internal investigations of the dynamic response

for different tube lengths, it is estimated that the accuracy without any dynamic reconstruction is fine for frequencies up to 5 -10 Hz. The statistical uncertainty is finally considered insignificant considering the number of data points.

The identified components with an estimated significant contribution to the accuracy of the results include the accuracy of the Pitot mounting angle and the upwash correction. The accuracy of the Pitot mounting angle is empirically estimated to be as high as $\pm 2\,\mathrm{deg}$. The impact of a potential offset of the rotor curve $c_l$ relative to the wind tunnel curve $c_l$ vs. $AoA$ is

attributed mainly to this uncertainty and not to a physical mechanism or a phenomenon of the rotor aerodynamics. For the current analysis, the curve $c_l$ was shifted by +2.5 deg to match the zero-lift angle of attack of the airfoil with the flap in its non-actuated position. Measuring the exact mounting angle of the Pitot probe with respect to the local chord of the rotor blade proved to be a complex undertaking, as there is no fixed reference for the measurement and one has to rely fully on the assumed CAD compatibility of the flyboard and blade leading edge shape.

Concerning the impact of uncertainty on the up-wash correction, this is more complicated. The main impact of an inaccurate upwash correction will be a different slope of the rotor curve $c_l$ versus AoA compared to the wind tunnel data. However, for a



| | Phase 4 |
|---|---|
| Date | July 2021 - Aug 2022 |
| Turbine | SG-4.3-120 DD |
| AFS revision | FT008rev10 |
| AFS actuation | continuously adjustable |
| Validation type | on-off, cyclic 1P |
| Location on blade | 38.0 - 58.0 m |
| Other tests | Wake rake, pressure belt, Pitot probe |

**Table 1.** Campaign information

blade section with flap actuation this is even more complex, as the bound circulation can be changed both due to an angle of attack change and/or flap actuation. One of the best procedures to reduce this uncertainty is to carry out 2D CFD blade sectional computations and extract the inflow velocity vector at the same position as the Pitot tube for the different activations level of

the flap. This is the procedure used in the present post-processing as discussed above, but other methods using full 3D CFD rotor computations as the basis have been proposed in the past, e.g. (Johansen and Sørensen, 2004; Shen et al., 2006; Madsen and Fischer, 2009). A comparison with these other approaches could lead to an improved insight into the upwash correction and reduce the uncertainty of the AoA measurement.

**3   Measurement campaign**

Excluding installation and demounting time, the test campaign extended over a period of approximately 24 hours from 17:30 on 28-07-2022 to 17:00 on 29-07-2022. These particular tests were part of the final phase 4 of the VIAs project testing, which was extended from July 2021 to August 22 (see Table 1). During these two days, several individual tests were performed, mainly in which the operating conditions of the AFS were varied. The turbine was operating mainly below rated conditions during the

entire duration of the test, with wind speeds in the range of 3 to 10 m / s and rotor speed in the range of approximately 6 to 10 rpm, corresponding to chord-based Reynolds numbers in the range of $2.1 - 3.6 \times 10^6$. One of the low-practical learnings of this campaign was that it is difficult to obtain measurements at high wind speeds with similar setups. The reason for this is the conflict with the required installation and demounting weather windows, which require rather low wind speeds for the safe operation of installation cranes. For future measurements, the goal is to design a more weather-resistant system such that the

instrumentation can be left on the turbine for a longer period, removing restrictions on the installation weather windows.



# 4 Results

The complete measurement period consists, as mentioned, of individual tests in which the active flap was actuated in different sequences. Furthermore, since this measurement technique with both the pressure belt and wake rake was tested for the first time, some proof-of-concept tests were performed, such as applying zig-zag tape to the leading edge of the airfoil to assess if the wake rake system has enough fidelity to measure such local changes in the rotating and turbulent environment.

As a first exemplary result, some characteristic time traces of different calculated variables are shown in Fig. 9. In Fig. 9a, an example of block averaging and moving window averaging is shown for a short measurement sequence of 60 s. Effectively, this averaging works as a very simple low-pass filter to avoid high-frequency peaks in the measured signals, making the analysis of the results clearer. The main frequency content of interest of the signals, which is around 1P, is far from the time-scale of the averaging windows and therefore unaffected by it. It is important to maintain the averaging window narrow to avoid introducing any artificial phase shift of the signal. With the averaging parameters chosen by the authors, the averaging procedure of the wake rake and flyboard signals corresponds to a down-sampling from 100 Hz to 25 Hz in an average-preserving manner.

Figure 9b shows an example of the angle of attack, as well as the lift and drag coefficients for the airfoil section during a stepwise flap actuation test. For clarity, the flap actuation pressure is shown in each graph as a visualization aid. This particular example is chosen to display the ability of the system to capture the shift in aerodynamic properties caused by the actuation of the active flap. As seen in the figure, shortly after 10:36 a flap actuation cycle begins, swapping between on/off activation every 60 s. The angle of attack is only minor affected by the flap actuation, whereas there is a clear impact on the lift and drag coefficients.

However, it should be noted that for this particular test case the VG's were removed in the spanwise region of the pressure belt and wake rake, and the blade surface was cleaned. The fact that the VG's were removed might explain some of the strong increase in drag when the flap was activated, probably causing some local separation over the suction side of the flaps. The period of the activation cycles for this particular case is shown to ensure that several rotor revolutions are captured per activation cycle, such that the 1P and active flap effects can be clearly identified.

The plots of Fig. 9 are for qualitative illustration only. For quantitative analysis longer periods are required. For this purpose, the full measurement campaign is divided into individual tests, where the activation level (i.e. the pressure level) of the AFS and the actuation frequency are varied. Some of these individual tests have a duration of approximately 10 minutes, while others have a duration of several hours.

As a first example, a test in which the AFS was actuated at constant time intervals of (60 s on, 60 s off) is shown in Fig. 10. The results are presented as airfoil polars that have been filtered for two individual pressure levels. The two levels of this figure are labeled *AFS inactive* and *AFS active*, corresponding to the flap being fully retracted, and activated with a control set point of 75 %, respectively. At this control set point, the flap reaches an angular deflection of approximately $10\,\mathrm{deg}$. From the lift polar, the impact of AFS activation is clearly visible. At a representative angle of 6.5 deg a variation of the lift coefficient of $\Delta c_l = 0.26$ is measured. For the same angle, the wind tunnel data suggest $\Delta c_{l,\mathrm{WT}} = 0.28$. For clarification, note that the wind tunnel data plotted in Fig. 10 are based on deltas applied to a reference airfoil measured in the wind tunnel. However, because





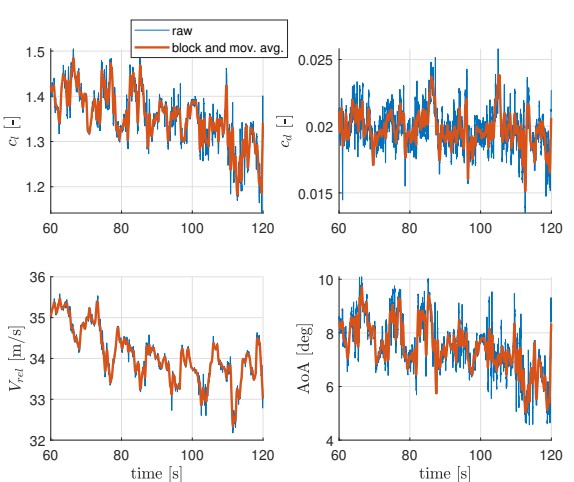

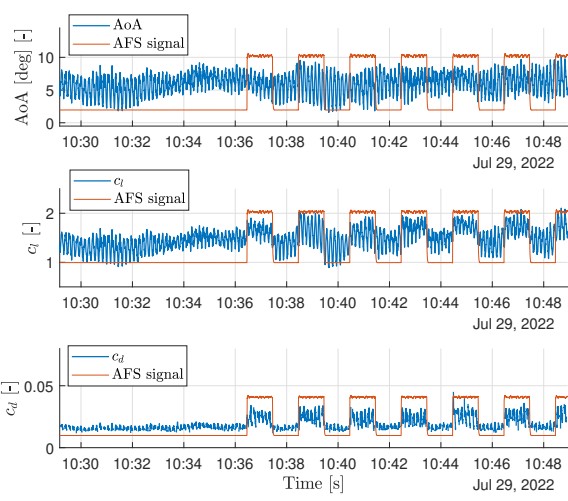

(a) Exemplary time traces of calculated variables over 60 s with and without block and moving average

(b) Exemplary time traces of AoA, $c_l$, and $c_d$. The activation signal is plotted for visualization purpose only.

**Figure 9.** Exemplary time traces – left figure shows the impact of the block averaging and moving averaging, right figure shows an example of the processed signals for step actuation of the active flap, where the pressure signal is plotted in the background with an arbitrary scaling for the purpose of visualization. For this particular test case the VG's were removed in the spanwise region of the pressure belt and wake rake measurements and the blade surface cleaned.

the airfoil geometry measured in the wind tunnel and the flap-chord scaling do not match 1:1 (owing to the maximum chord limits of the wind tunnel), the AFS deltas due to actuation in the tunnel are *scaled down* to match the real flap length to chord ratio at the actual station at which the pressure belt was installed.

    The remaining polar plots for the drag coefficient and the lift-to-drag ratio show very good agreement between the field measurements and the wind tunnel measurements when considering the deltas due to AFS activation. The absolute levels of

drag measured with the wake rake are observed to be above the drag measured in the wind tunnel. Whether this is correct or not is a matter for future investigation, e.g. if the turbulent outer flow causes a higher drag level. However, the trends of both the drag and the lift ratio as a function of the angle of attack correlate well.

    As a further observation, it is seen that this version of the active flap leads to a visible increase in the drag coefficient base level. During the wind tunnel tests of this flap version, it was identified that the velocity deficit width behind the flap increases

with the level of flap deflection. Although this result is not surprising, it highlights the need to improve the pressure recovery profile on the aft portion of the airfoil and its transition to the active flap. To better understand the source of the drag increase and to be able to correlate it with the velocity deficit measurements of the wake rake and the pressure belt, the signals of these two instruments were filtered for a narrow range of angle of attack in the interval $6 \pm 0.2\,\mathrm{deg}$. Both the pressure distribution and the wake deficit measurements are shown in Fig. 11. Note in particular two characteristics of the wake rake velocity deficit



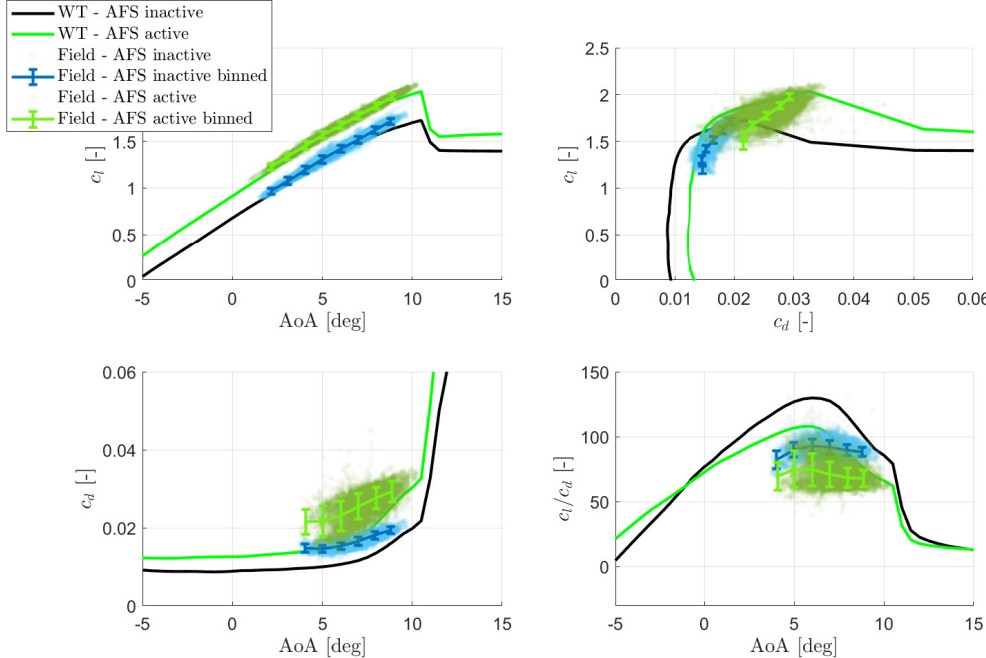

**Figure 10.** Exemplary results of polars measured for the airfoil with and without active flap actuation

with and without AFS actuation: a shift toward lower $y-$ values, i.e. a downward deflection of the airfoil wake due to the flap deflection, and an asymmetric widening of the velocity deficit showing a wider deficit on the suction side (toward positive $y-$ values) due to the local thickening of the boundary layer and small recirculation region directly behind the flap.

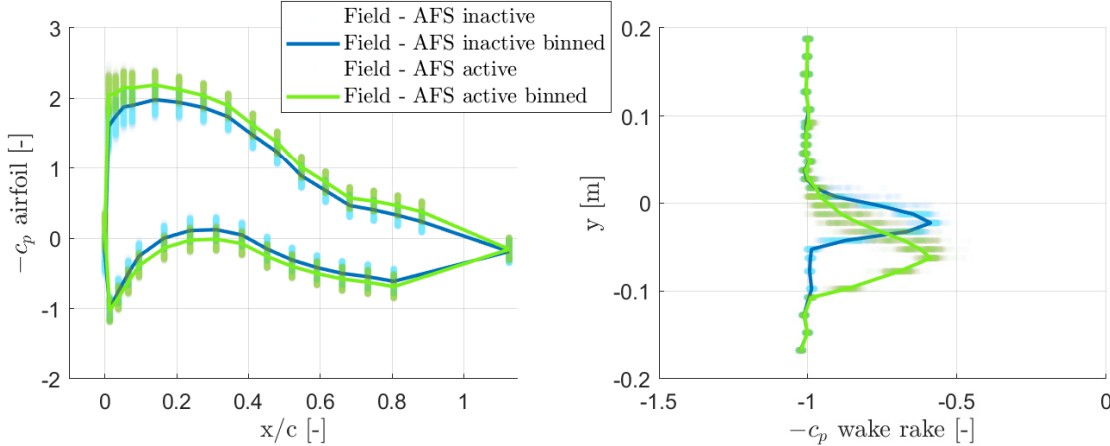

**Figure 11.** Exemplary results of $c_p$ distributions and wake deficits for the airfoil with and without active flap actuation for the range of $\alpha = 6 \pm 0.2 \, \mathrm{deg}$.





As a further demonstration example, an individual test is shown in which a 0.4 mm thick zigzag tape (ZZ) was applied in a location close to 2 % of the airfoil chord on the suction side. Similarly as for the previous example, the airfoil polars are

shown separately for the case of the airfoil with and without ZZ tape. For this case, the main focus is on the impact on the drag increase. The airfoil polars with and without ZZ tape, as well as the $c_p$ plots for the pressure belt and the wake rake for $\alpha = 6 \pm 0.2 \, \mathrm{deg}$ are shown in Figs. 12 and 13, respectively. Interestingly, the impact on the lift coefficient after the application of the ZZ tape is negligible, which is not the experience from wind tunnel measurements which usually show a small penalty in the mean lift level. If this is true, or just an artifact of this particular measurement, it remains unanswered and is subject of

future investigations.

The impact on the drag coefficient is quite clear. The drag level is seen to be higher in its mean level, but also starts to deviate towards higher $c_d$ values earlier compared to the clean airfoil. Compared to wake-rake field measurements against wind-tunnel measurements, the drag increase as a function of angle of attack is seen to occur in a more gradual manner in the field, in contrast to a sharper increase prior to entering stall, as observed in the wind tunnel. This difference could be attributed to the

higher level of outer turbulence in the field that leads to a faster boundary layer development because of the outer flow bypass transition mechanisms. However, this is only a hypothesis.

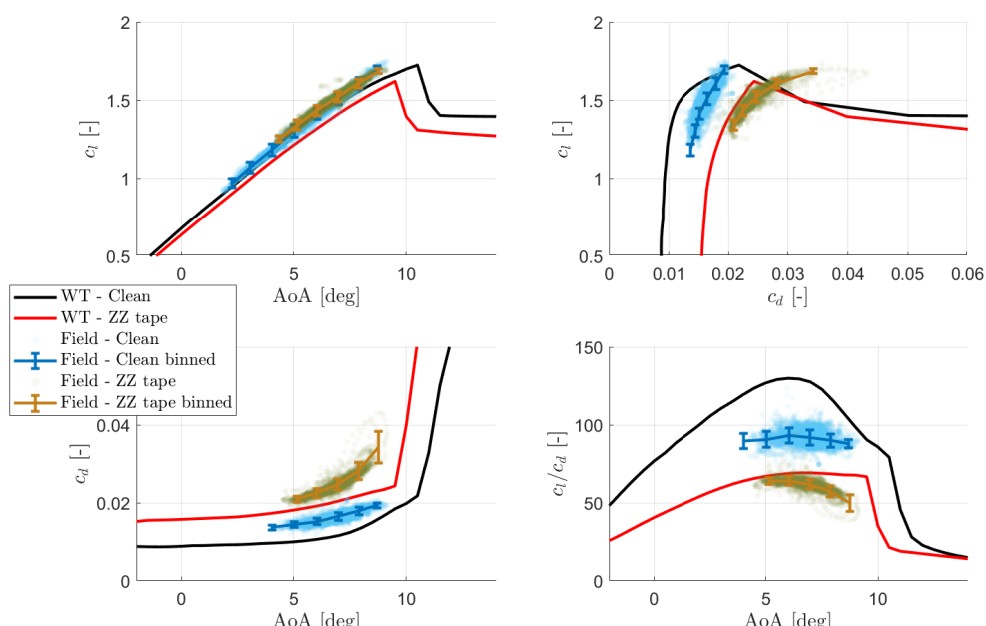

**Figure 12.** Exemplary results of polars measured for the airfoil with and without zig-zag tape

The comparison of the pressure belt and the wake rake pressure coefficients is shown in Fig. 13, where it is worth highlighting the differences in the velocity deficit profile. For the test with ZZ tape, the velocity deficit is wider toward the suction side due




to the thickening of the boundary layer caused by the ZZ tape. Furthermore, the velocity deficit is deeper, a sign of the overall

higher viscous losses of the flow.

If one compares the velocity deficit depth of the airfoil section with the AFS actuation in Fig. 11 with the ZZ tape deficit, it becomes clear that the increase in drag in the case of the ZZ tape is attributed to higher viscous losses (predominantly deeper symmetric deficit), while the increase in drag for the case of the AFS in its activated position is predominantly due to increased pressure drag in the trailing edge area (predominantly wider asymmetric velocity deficit).

Being able to draw this type of conclusions from field measurements highlights how promising these types of measurements are with regard to closing the gap of understanding between airfoils in the field vs. airfoils in the wind tunnel.

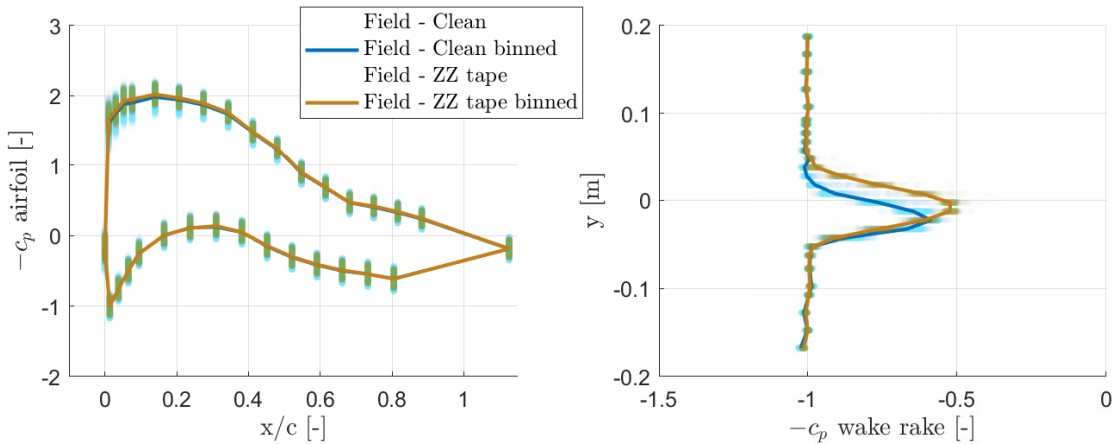

**Figure 13.** Exemplary results of $c_p$ distributions and wake deficits for the airfoil with and without zig-zag tape

## 5 Discussion and outlook

The present measurement campaign has been a kind of proof of concept for measuring the airfoil section drag under natural operating conditions of a full-scale turbine. It is believed to be the first of its kind, where simultaneous measurement of airfoil

pressure distribution, inflow angle and velocity, as well as wake velocity deficit, allows the characterization of $c_l$, $c_d$, and $c_p$ in a turbulent and rotating environment. Instrumentation and associated processing methods were developed to allow for detailed field characterization of an active flap system. The results of this measurement campaign provide valuable insight into the performance of the active flap system. However, the potential use cases of this novel measurement technique cover a wide range of aerodynamic applications, e.g. the study of airfoil characteristics at very high Reynolds number or the impact of

atmospheric turbulence on natural transition and drag, to name just two examples.

It is seen that the system is able to measure small differences in the aerodynamic state of an airfoil section, e.g. drag differences from a clean leading edge vs. an artificially tripped boundary layer. With modern instrumentation and data acquisition systems with high accuracy in the analogue-digital interface, the main uncertainty in this type of measurements does not come





from the instruments or from the statistical uncertainty, but rather from the overall corrections required, e.g. the upwash correction and the measurement of the free flow dynamic pressure. This is particularly true for rotor blades with active flow control devices such as active flaps which influence the total circulation level, such that the upwash at the probe location is not only a function of the angle of attack, but also a function of the actuation level of the device.

Some items remain unanswered or with the potential for future improvement such as the correct approach for the upwash correction for the Pitot probe (e.g. via full 3D rotor CFD), an improved measurement of the physical alignment of the system with the local chord blade (e.g. via photogrammetry), considerations related to spanwise drag variations due to aerodynamic add-ons such as serrations, and the understanding of the high absolute drag level. Furthermore, with the learning's from this first campaign, several improvement areas were identified for the hardware system itself, including:

– Higher weather robustness to allow for longer measurement campaigns

– Improved mounting alignment procedure and measurement of mounting angles

– Improved blade mounting strategy and procedure (potentially more compact system)

– Improved spanwise stiffness of the mounting frame

In general, this campaign has shown that additional measurement of airfoil section drag, in combination with the measurement of the pressure distribution, opens a range of promising new options for optimizing airfoil performance in its real operating environment. It could e.g. be optimizing size and position of vortex generators, either on new prototype blades or maybe on blades with some degradation of the surface.

## 6 Conclusions

The development and validation of a novel measurement system for full-scale drag measurements is presented. The system is developed in an effort to improve the understanding of rotor blade aerodynamics, with the aim of reducing the knowledge gap in airfoil sectional characteristics measured at wind tunnel and full-scale level. This measurement technique and its related instrumentation, using a combination of inflow measurements (Pitot tube), pressure distribution (pressure belt), and velocity deficit (wake-rake), allow the characterization of the aerodynamic state of the local sections to an unprecedented level of detail.

The performance of the full-scale measurement system is exemplified in a two-day campaign on a modern 4.3 MW direct drive wind turbine with a rotor blade equipped with an active flap system. The benefit of testing the system on such a blade is two-fold: first, it demonstrates the ability of the system to measure changes in the aerodynamic characteristics of the airfoil when activating the flap in different modes and secondly, the data collected support the efforts of the full-scale aeroelastic characterization of the active flap system.

The capabilities of the system are shown with two exemplary tests within the same measurement campaign, the first in which the active flap system is actuated in prescribed intervals, and the second in which the airfoil section was measured with and without ZZ tape. The former is performed to demonstrate the ability of the system to measure changes in both lift and drag



related to changes in circulation and to compare this with wind tunnel measurements. The latter test is performed to demonstrate the fidelity of the wake-rake system in capturing local modifications of the boundary layer characteristics due to changes in viscous drag. In both cases, demonstrated in a rotating and turbulent environment, the system performed as expected.

However, from the measurements collected and the comparison with wind tunnel tests, it is seen that the absolute drag levels measured on the rotor are considerably higher. This possible difference in drag levels in the wind tunnel and on the rotor is exactly what we want to study with the new instrumentation. Further investigations of the present data set and new measurement campaigns will contribute to understanding what the true drag levels are. The deviations we see in the present data set could be related to the uncertainty of dynamic pressure measurement and will be an area of improvement for future tests. Also fundamental issues to extract the drag from a highly unsteady velocity deficit is a research area.

Ultimately, the continued development and optimization of blade-mounted aerodynamic measurement systems such as the one presented herein will enable the closing of knowledge gaps of the aerodynamic aerodynamic characteristics of rotor blades in full-scale environments.

*Data availability.* Due to confidentiality reasons, the data is only available to the project partners.

*Video supplement.* A short video clip of the wake rake test can be found under https://www.linkedin.com/posts/helge-aagaard-madsen-ba006b93_demonstration-of-an-autonomous-measurement-activity-6970317822740570112--azA?utm_source=share&utm_medium=member_desktop&rcm=ACoAABPDpugB4LCfu2ALZyEjKEWnzV_xptui4xc

*Author contributions.* All authors contributed to the work presented in this research paper. HAaM contributed in the planning of the test, the data evaluation and drafting of the article, AGG contributed with the planning and execution of the test, the design of the turbine attachment concept, data analysis and drafting of the paper, TB contributed with the planning of the test and writing a major part of the data processing software, ASO and AF contributed with the wake rake design and test on the rotating test rig and with the data processing algorithms for the data collected by the wake rake and the flyboard, SBI contributed with the design of the wake rake and the execution of the test.

*Competing interests.* The authors declare no competing interest in the contents of this article.

*Acknowledgements.* We acknowledge funding from EUDP-2019, journal nr. 64019-0061 of the VIAs project making it possible to carry out the tests. Further data analysis has been partially funded by EUDP journal nr. 640241-521566 of the FAR project.



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
