# Peer review of "Blade surface pressure and drag measurement of a blade section on a 4.3 MW turbine with trailing edge flaps"

_Wind Energy Science, 2025_

## Referee Comment (RC3)

**Review paper WES entitled :**

**Blade surface pressure and drag measurement of a blade section on a 4.3 MW turbine with trailing edge flaps**

from :

Helge Aagaard Madsen, Alejandro Gomez Gonzalez, Thanasis Barlas, Anders Smærup Olsen, Sigurd Brabæk Ildvedsen, and Andreas Fischer

**Resume :**

Authors present an original way of measuring drag on field test measurements. This can be used to evaluate drag with and without a control/perturbation device (trailing edge flaps, Vgs, zztape). This method is well known for wind tunnel tests, but has not been used yet on field tests.

Evaluating efficiency of control or perturbation device on field tests is an important challenge. These measurements are thus rare and essential to help modeling aerodynamics in real operating conditions. This is also really complex measurements to perform. This study is therefore highly valuable. However, there is many major issues that needs to be tackled before publication.

**Summary of major issues :**

The paper present an important work with wind tunnel, tests ring and field test measurements, that are however most of the time not sufficiently described, not sufficiently justified and analyzed. This is also difficult to make a link between wind tunnel tests/ test ring and wind turbine tests as there is either no quantitative results, only the Cl/Cd curve or only the Cl/Cd time series ... I recommand authors to maybe focus more on one set of tests and to provide a more detailed analysis.

**Major issues : some details**

**1-Introduction :**

Authors underline the lack of knowledge on blade aerodynamics on solely three mechanisms : high Reynolds numbers, roughness state at the surface of the blades, outer-flow state (turbulence inflow) that acts on the laminar to turbulent transition on the airfoil surface.

However, differences between wind tunnel and field tests can come from more than these 3 mechanisms. Indeed, wind tunnel turbulent inflow do not necessarily match inflows encountered in blade real operating conditions and not only regarding turbulence level. Mean inflow velocity gradients combined with different turbulent inflows may lead to different operating AoA. This can induce important load variations and even flow separation (even small) which induces 3D and unsteady flow over the airfoil and thus load variations. When the flow separates other aspects such as the airfoil aspect ratio, and the centrifugal forces play an important role in differences with wind tunnel measurements. Centrifugal forces are reproduced by the rotating test ring, however, the aspect ratio is very low (2) on both wind tunnel and rotating test ring.

Please reformulate hypothesis of this study in the introduction :

- The author place themselves in the hypothesis with no inflow inhomogeneity (mean or turbulent) along the blade in real operation (there is no inflow measurements able to measure the presence of any mean or unsteady velocity gradient). Only two points are measured in the near blade inflow (in the blade induction zone?) using the 5 hole pressure sensors.

- The AoA is sufficiently low for the flow to stay attached.

**2 - Experimental set-up :**

• Figure 1 is unclear. It is very difficult to see how the rake is installed relatively to the blade.

Please provide a clearer view (maybe add a sketch).

• Due to all the instrumentation (wake rake plus 5 hole pressure sensors), the flow over the airfoil might be 3D even if not separated (low AoA). This could happen especially due to the low aspect ratio (equal to 2).

**How the two dimensionality is checked (wood tuft ? Visualization??)?**

• The wake rake composition is not given clearly in section 2.2. A picture appears only in L120, with no explanation on the location of « static tubes », « total pressure tubes » or « head of pitot tubes » that seems to compose the wake rake as given from L116 to L119.

**Please provide a clear description/composition of the wake rake (on figure 2), at section 2.2. Please justify the choice of this composition.**

The geometry of the airfoil section is not given. This is a very important matter as the state of the flow over the airfoil depends largely on the airfoil geometry. 15° is certainly a separated state, and depending on the airfoil geometry (thickness, curvature), this correspond to a deep stall or a slight trailing edge flow separation ...
Also, there is no means to evaluate the drag without the wake rake (no reference case). This could be a an xfoil simulation (valid only at low AoA) or drag measurements using a balance system.

**Please provide the airfoil geometry that is tested, and at least a reference case with measurement of Cl, Cd versus AoA.**

• L110-114 : « measurement is only slightly affected by ».

**How can you conclude on this as there is no « reference case »? Also, please provide a more quantitative description.**

• L113 « The cd is only slightly affected by the pressure orifices, but largely affected by the five-hole Pitot tube connection »

**This is to me a major issue for drag measurement. Please develop more for which configurations this sentence applies ? Please provide a quantification of this impact. At last, justify more why using 5 hole pressure sensors, and give possible solutions for future work.**

• Nothing is said on impact of AoA (8° and 15° were tested) on drag measurements

Please provide Cd error quantification for both AoA.

**Section 2.2.2**

• L136 « Detailed tests were carried out in different wind conditions ... »

*Please provide detailed informations on tested configurations (give values of velocity, pitch and flap angles, together with calculated AoA – and provide method used to compute AoA).*

• P6 L137 : There is no description of the VG and of the surface roughness (shape, location on the blade, wind tunnel tests etc ...) used for figure 4. Also this figure is out of the main paper subject.

*Either the authors give more information on these tests, or the figure should be removed.*

Also, why not providing a comparison between wind tunnel drag measurements and test ring measurements (and then field tests measurements) ? Isn't is the purpose of the paper ?

**Section 2.2.3**

• L144 « it was decided to design and manufacture a new frame for attachement of the wake rake on the full scale blade »

I do understand this difficulty, however, do you have reproduced this set-up in wind tunnel (at least once) to evaluate this impact of this new mounting system ? If not, how do you expect this mounting to change results ? Please give more details.

**Minor comments :**

• L21-22 : « This is due to ... 2D wind tunnel flow and unsteady, turbulent 3D flow experienced on the wind turbine blade »

I agree that there exist major differences from wind tunnel experiments and field measurements, but I won't say that the inflow is 2D in wind tunnels. In wind tunnels the flow is homogeneous and with low turbulence intensity, unless customized differently (passive or active grids, gust or other generated perturbations ...).

please be more specific

• L54 : « This is commonly measured ... »

This is not clear what « This » refers to. One could think that the wake rake measure only the viscous drag.

**Please reformulate.**

• « Experimental set-up » section :

The description of experimental set-up is made of bullet points, which is unusual and rather adapted to technical reports.

Please provide sentences rather than bullet points.

• P6L133 : « in standar procedure from inside »

Please develop.

---

## Author Comment (AC1)

**Comments to Reviewer #1**

Dear Reviewer

First of all we would like to thank you for your constructive review of our article. Please find below our responses (in red) to your comments (in blue) and article updated text (in brown). Please find also attached a marked-up version showing all changes in the article.

Yours sincerely

H. Aa. Madsen and co-authors

This is a very interesting paper with a unique measurement on a full scale wt. I have only three small comments/recommendations:

Thank you for your review of the article and comments.

1) In Fig. 6b the green tape is hardly visible. It would be nice if a close-up photo could be included. A close-up photo of the green tape on the serrations has been included as figure 4c (clarification comment, the figure has now number "4" instead of "6").

2) If any turbulence data from the met mats are available they should be mention in chapter 3. Information for the met-mast measurements is provided in the references, however, no met-mast data is used for the analysis pertinent to this article, therefore no further details are given. Note also the following text of the article in the "Turbine Instrumentation" subsection: "...as the met-mast used specifically for phase 4 of the test was no longer in operation the days when the wake-rake test was conducted."

3) The existence of the VGs should be mentioned already in section 2.3.2 Text mentioning the VGs is added in 2.3.2. Furthermore, it is explained why the VGs were removed for this specific test with the following text addition:

Furthermore, the blade has vortex generators (VGs) mounted on the suction side. However, these VGs were removed prior to the execution of the test in question as discussed in section 2.3.2 to avoid spanwise variations of the wake velocity deficit coming from the upwash and downwash structures trailed from the VG vanes.

14-07-2025 13:29:27

**Compare Results**

Old File:

**WES\_paper\_2025\_75\_1st\_version.pdf 24 pages (9.71 MB)**

28-04-2025 07:33:32

versus

New File:

WES\_paper\_2025\_75\_reviewed.pdf

23 pages (7.38 MB) 14-07-2025 11:24:43

Total Changes

296

Text only comparison

Content

118 Replacements 78

100 Insertions

Deletions

Styling and Annotations

0 Styling

 $\mathbf{O}$ Annotations

Go to First Change (page 1)

**Blade surface pressure and drag measurement of a blade section on a 4.3 MW turbine with trailing edge flaps**

Helge Aagaard Madsen1, Alejandro Gomez Gonzalez2, Thanasis Barlas1, Anders Smærup Olsen1, Sigurd Brabæk Ildvedsen1, and Andreas Fischer1

1Department of Wind and Energy Systems, Technical University of Denmark, 4000 Roskilde, Denmark 2Technology Development, Siemens Gamesa Renewable Energy, 7330 Brande, Denmark.

Correspondence: Helge Aagaard Madsen (hama@dtu.dk)

[revised manuscript text omitted]

**2.2.1 Prototype design and wind tunnel tunnel testing**

Wind tunnel tests were conducted in the PLCT to estimate the main dimensions of the wake rake design. However, it should be noted that the design was also based on the already developed wake rake installed as part of the standard measurement system in the PLC wind tunnel at DTU.

The test section in the PLCT is 2 m in span and 3 m width. For this campaign, a blade section with a 1 m chord and a 2 m span was used. The blade section was equipped with a trailing edge morphing flap, as one of the crucial design parameters

|                    | Wake rake wind tunnel tests                                           |
|--------------------|-----------------------------------------------------------------------|
| Tunnel flow speed  | 20 m/s (Re = $1.3 \times 10^6$ ) and 30 m/s (Re = $1.9 \times 10^6$ ) |
| Distance behind TE | 0.1 m, 0.3 m, 0.5 m and 1.0 m                                         |
| Vertical position  | 1.0 m, 1.15 m and 1.38 m                                              |
| Angle of attack    | 8 and 15 deg                                                          |
| Flap deflection    | 0 mm, -20 mm and 20 mm                                                |

**Table 1.** Overview of the parameters varied during the wind tunnel tests forming the basis for the final wake rake design. The three vertical positions were chosen to be: 1) behind pressure orifices (1.0 m); 2) slightly above pressure orifices (1.15 m) and 3) maximum height, closer to the connecting tubes for the five-hole Pitot tube (1.38 m).

to determine in the PLCT tests is the width of the wake rake so it can capture the deficit with the flap in its maximum and minimum deflection positions. The instrumentation comprised standard pressure orifices in the midspan and two five-hole Pitot
tubes at 0.5-m and 1.5-m span positions measured from the floor of the tunnel.

The parameters varied during the wind tunnel tests are listed in Table 1. The test parameters in the wind tunnel were chosen to match conditions similar to the wake rake test on the rotating test rig.

The drag coefficient  $c_d$  is found using the method of Jones (1936) and may be expressed as:

$$c_d = 2 \int_{y_{min}}^{y_{max}} \sqrt{c_{p,tot} - c_{p,stat}} \left[ 1 - \sqrt{c_{p,tot}} \right] dy, \tag{1}$$

95 with

$$c_{p,tot} = \frac{H_0 - p_0}{0.5\rho U_0^2}, c_{p,stat} = \frac{p - p_0}{0.5\rho U_0^2},\tag{2}$$

Solution where  $y_{\text{max}} - y_{\text{min}}$  is the length of the integration path of the wake rake velocity deficit,  $p_0$  the static pressure, p the dynamic pressure,  $H_0$  the total head,  $\rho$  the density, and  $U_0$  the velocity of the free stream.

However, we use the same formula for the derivation of the drag coefficient in turbulent flow, and the accuracy of this needs to be further investigated in future tests.

From the wind tunnel test results, the following conclusions and parameter settings were derived: 1) The  $c_d$  measurement is only slightly affected by the downstream distance from the trailing edge (0.1 m to 0.5 m); 2) The  $c_d$  measurement is only slightly affected by rotating the rake with the profile, at least not for the value of angle of attack of 8 deg (It is also a matter of making the end of the tube conical); 3) The  $c_d$  is only slightly affected by the pressure orifices, but largely affected by the

five-hole Pitot tube connection and 4) The wake width is approximately 0.3 m at  $\alpha = 8$  deg for a flap deflection of  $\delta = \pm 20$  mm.

|                                     | Final wake rake dimensions                           |
|-------------------------------------|------------------------------------------------------|
| Position of the head of Pitot tubes | 0.3 m behind the TE                                  |
| Wake rake width                     | 0.4 m (0.1 m wider than the width measured in tests) |
| Static tubes                        | 4 in line with the total pressure tubes              |
| Total head tubes                    | 56 with 5 mm and 10 mm spacing, respectively         |

Table 2. Overview of the final wake rake design parameters. See also the photo of the wake rake in Fig.1.

Based on the findings and conclusions of the wind tunnel tests, the final design parameters as shown in Fig. 2 were chosen. A photo of the manufactured wake rake is shown in Fig. 1. Inside the wake rake plastic tubes are connected to the pressure probes and led out at both ends of the rake to the pressure scanners outside.

Figure 1. Photo of the final manufactured wake rake. See also the parameters and dimensions in Table 2.

**110 2.2.2 Mid scale testing of the functionality and integration of the wake rake on the blade section on a rotating test rig**

To gain experience with the functionality and operation of the wake rake in a rotating environment before going to the full scale turbine application, the wake rake was installed and tested on a rotating test rig (RTR) Madsen et al. (2015), Ai et al. (2019) and Barlas et al. (2022). The RTR is built on a 100 kW turbine platform where the rotor is replaced by a 10 m pitch-able long boom carrying a 2 m span blade section at the tip, see the right photo in Fig. 2. In fact, a blade section for tests in the PLC

115 tunnel can be tested directly on the RTR.

The wake rake was mounted on the blade section with a frame of circular carbon tubes attached to the aluminum frame inside the blade section; see the photos in Fig. 2. The 2 m long rectangular blade section + end caps has an active trailing edge flap system with serrations like in the full scale experiment. However, the serrations were covered with tape as it is expected that the flow will not be uniform in the position of the wake rake due to the serrations and thus make the drag measurement uncertain. The wake rake probes were connected to four 16 channel differential pressure scanners positioned inside the blade

Ø

120 uncertain. The wake rake probes were connected to four 16 channel differential pressure scanners positioned inside the blade section and with the reference port connected to the static port of one of the five hole pitot tubes attached to the leading edge of the blade section.

The RTR tests confirmed that the wake deficit of the blade section could be satisfactorily covered within the wake rake width even in the turbulent inflow and with flap activation. The tests also confirmed that the carrying frame was stable.

---

## Author Comment (AC2)

**Comments to Reviewer #2**
Dear Reviewer

First of all we would like to thank you for your constructive review of our article. Please find below our responses (in red) to your comments (in blue) and article updated text (in brown). Please find also attached a marked-up version showing all changes in the article.

Yours sincerely

H. Aa. Madsen and co-authors

General comments: A very nice article about the evolution of an innovative measurement technique that contributes to closing the gap between the wind tunnel results used in design and unknown sectional performance in the field. From this viewpoint the article provides new and valuable insight and is worthy of publication. The storyline is clear and well conveyed. It is recommended to follow up on the below suggestions for improvement:
Thank you for your general comments.

-p4/5 section 2.2.1 Can the authors comment on the applicability of Jones' method in a rotor setting? In particular concerning static pressure assumptions made by this method and the static pressure variations that might be present due to rotor blockage.
If one imagines a control volume surrounding the local airfoil, then Jones' method should be applicable irrespective of the rotor blockage, this is because the static pressure is being measured also in the rotor plane (and not at an undisturbed upstream position). However, Jones' method is generally not proved to be applicable to unsteady 3D flows and the static pressure measured at the wake rake will also vary during a rotor revolution. To highlight this uncertainty, we have inserted the following text in the paper:
However, we use the same formula for the derivation of the drag coefficient in turbulent flow, and the accuracy of this needs to be further investigated in future tests.

-p5 section 2.2.1 Some illustration of the mentioned results from the parameter variations are recommended. In addition to that, has the influence of the downstream position of the wake rake on the result also been verified?
The influence of the downstream position of the wake rake has been investigated in the wind tunnel as mentioned in 2.2.1 but not on the rotor.
See line 110 on page 5 of the manuscript: "– The cd measurement is only slightly affected by the downstream distance from the trailing edge (0.1 m to 0.5 m)

-p12 section 2.3.6 and section 4 The block and moving averaging procedure is explained to act as a low pass filter. In the end the results are presented by means of sectional polar plots like fig.10, which appear quite steady without hysteresis. In how far does the post-processing procedure filter out the dynamics present in the field? What is the non-dimensional frequency k at this section based on 1P? Would it be interesting to also show a polar plot like figure 10 without the pre-processing procedure (based on raw data),or would that result in a rather large cloud op

points?

No block and moving averaging would result in large scatter in the polars and the time-series plots becomes unclear. Since the interest lies in comparison of 'static' polars, the method is considered adequate. The estimated reduced frequency is low : k= omega*0.5c/Vrel, about 0.009 for Vrel = 40m/s and omega =0.73 (quasi-steady assumtion valid generally for k<0.05).

-p13 section 2.3.7 Uncertainty Pressure measurements on wind turbines are often complicated by drift due to, e.g. temperature changes and the difficulty of a providing a stable reference pressure. Although there is a referral to a previous paper from 2022 about this measurement system, could the authors comment on these challenges in relation to their system?

No known drift due to temperature changes has been observed. The reference pressure is taken as the static pressure from the wake rake. The inflow/belt pressures are differential pressures, in the case of absolute pressure, care must be taken to compensate both for barometric pressure changes as well as hydro-static, but this is fortunately not the case for this experiment.

Furthermore, from the 16 channel miniature Evoscanner data sheet the following information is given regarding temperature compensation: *"Integrated temperature sensors provide useful data, but also apply temperature correction to every pressure sensor, at source, to ensure optimal performance and minimal ambient temperature effects."*

-p16 section 4 A discrepancy in the drag coefficient between field and wind tunnel is noted. Is the same offset present between RTR and wind tunnel?

There was also an offset in Cd between WT/RTR. In the full scale case there is also a pressure gradient due to centrifugal effects, so small flap angle difference to wind tunnel which is partially responsible for the Cd offset. In both cases, the difference can probably be attributed to field test flow and surface conditions.

However it should be noted that due to the comments from Reviewer #3 the measured airfoil characteristics from the RTR have been left out to focus fully on the difference between wind tunnel and full-scale airfoil characteristics.

-It is mentioned that the turbine has been instrumented to measure loads and power. It would be helpful if the influence of the added external sectional instrumentation on the loads and power can be quantified. Also, have the effects of flaps and zz-tape on power and loads also been measured or were these effects too small?

The effects on loads and power of the instrumentation on the power and load levels of the turbine are minimal and such differences would not be quantifiable unless a medium-term duration side-by-side test with a similar machine would be conducted, similar as when small changes in blade configurations are performed, e.g. when installing add-ons. The following text has been added describing that.

The impact of the installed instrumentation of the blade is expected to have a minimal impact on the operation on the turbine..

**Compare Results**

Old File:

**WES_paper_2025_75_1st_version.pdf**

24 pages (9.71 MB)

28-04-2025 07:33:32

versus

New File:

**WES_paper_2025_75_reviewed.pdf**

23 pages (7.38 MB)

14-07-2025 11:24:43

**Total Changes**

**296**

Text only comparison

**Content**

118 Replacements

100 Insertions

78 Deletions

**Styling and Annotations**

0 Styling

0 Annotations

Go to First Change (page 1)

[revised manuscript text omitted]

**2.2.1 Prototype design and wind tunnel tunnel testing**

Wind tunnel tests were conducted in the PLCT to estimate the main dimensions of the wake rake design. However, it should be noted that the design was also based on the already developed wake rake installed as part of the standard measurement system in the PLC wind tunnel at DTU.

The test section in the PLCT is 2 m in span and 3 m width. For this campaign, a blade section with a 1 m chord and a 2 m span was used. The blade section was equipped with a trailing edge morphing flap, as one of the crucial design parameters

| | Wake rake wind tunnel tests |
| --- | --- |
| Tunnel flow speed | 20 m/s (Re = $1.3 \times 10^6$) and 30 m/s (Re = $1.9 \times 10^6$) |
| Distance behind TE | 0.1 m, 0.3 m, 0.5 m and 1.0 m |
| Vertical position | 1.0 m, 1.15 m and 1.38 m |
| Angle of attack | 8 and 15 deg |
| Flap deflection | 0 mm, -20 mm and 20 mm |

**Table 1.** Overview of the parameters varied during the wind tunnel tests forming the basis for the final wake rake design. The three vertical positions were chosen to be: 1) behind pressure orifices (1.0 m); 2) slightly above pressure orifices (1.15 m) and 3) maximum height, closer to the connecting tubes for the five-hole Pitot tube (1.38 m).

to determine in the PLCT tests is the width of the wake rake so it can capture the deficit with the flap in its maximum and minimum deflection positions. The instrumentation comprised standard pressure orifices in the midspan and two five-hole Pitot tubes at 0.5-m and 1.5-m span positions measured from the floor of the tunnel.

The parameters varied during the wind tunnel tests are listed in Table 1. The test parameters in the wind tunnel were chosen to match conditions similar to the wake rake test on the rotating test rig.

The drag coefficient $c_d$ is found using the method of Jones (1936) and may be expressed as:

$$c_d = 2 \int_{y_{min}}^{y_{max}} \sqrt{c_{p,tot} - c_{p,stat}} \left[ 1 - \sqrt{c_{p,tot}} \right] dy,$$  (1)

with

$$c_{p,tot} = \frac{H_0 - p_0}{0.5\rho U_0^2}, c_{p,stat} = \frac{p - p_0}{0.5\rho U_0^2},$$  (2)

where $y_{max} - y_{min}$ is the length of the integration path of the wake rake velocity deficit, $p_0$ the static pressure, $p$ the dynamic pressure, $H_0$ the total head, $\rho$ the density, and $U_0$ the velocity of the free stream.

However, we use the same formula for the derivation of the drag coefficient in turbulent flow, and the accuracy of this needs to be further investigated in future tests.

From the wind tunnel test results, the following conclusions and parameter settings were derived: 1) The $c_d$ measurement is only slightly affected by the downstream distance from the trailing edge (0.1 m to 0.5 m); 2) The $c_d$ measurement is only slightly affected by rotating the rake with the profile, at least not for the value of angle of attack of 8 deg (It is also a matter of making the end of the tube conical); 3) The $c_d$ is only slightly affected by the pressure orifices, but largely affected by the five-hole Pitot tube connection and 4) The wake width is approximately 0.3 m at $\alpha = 8$ deg for a flap deflection of $\delta = \pm 20$ mm..

| | Final wake rake dimensions |
|---|---|
| Position of the head of Pitot tubes | 0.3 m behind the TE |
| Wake rake width | 0.4 m (0.1 m wider than the width measured in tests) |
| Static tubes | 4 in line with the total pressure tubes |
| Total head tubes | 56 with 5 mm and 10 mm spacing, respectively |

**Table 2.** Overview of the final wake rake design parameters. See also the photo of the wake rake in Fig.1.

Based on the findings and conclusions of the wind tunnel tests, the final design parameters as shown in Fig. 2 were chosen. A photo of the manufactured wake rake is shown in Fig. 1. Inside the wake rake plastic tubes are connected to the pressure probes and led out at both ends of the rake to the pressure scanners outside.

[Figure]

**Figure 1.** Photo of the final manufactured wake rake. See also the parameters and dimensions in Table 2.

**2.2.2 Mid scale testing of the functionality and integration of the wake rake on the blade section on a rotating test rig**

To gain experience with the functionality and operation of the wake rake in a rotating environment before going to the full scale turbine application, the wake rake was installed and tested on a rotating test rig (RTR) Madsen et al. (2015), Ai et al. (2019) and Barlas et al. (2022). The RTR is built on a 100 kW turbine platform where the rotor is replaced by a 10 m pitch-able long boom carrying a 2 m span blade section at the tip, see the right photo in Fig. 2. In fact, a blade section for tests in the PLC tunnel can be tested directly on the RTR.

The wake rake was mounted on the blade section with a frame of circular carbon tubes attached to the aluminum frame inside the blade section; see the photos in Fig. 2. The 2 m long rectangular blade section + end caps has an active trailing edge flap system with serrations like in the full scale experiment. However, the serrations were covered with tape as it is expected that the flow will not be uniform in the position of the wake rake due to the serrations and thus make the drag measurement uncertain. The wake rake probes were connected to four 16 channel differential pressure scanners positioned inside the blade section and with the reference port connected to the static port of one of the five hole pitot tubes attached the leading edge of the blade section.

The RTR tests confirmed that the wake deficit of the blade section could be satisfactorily covered within the wake rake width even in the turbulent inflow and with flap activation. The tests also confirmed that the carrying frame was stable. .

[revised manuscript text omitted]

In what follows a short description of the AFS is given, but the reader is referred to (Gomez Gonzalez et al., 2022, 2023) for further details. The outer portion of the blade (span location 38 m – 58 m) is retrofitted with the active flap (internal reference FT008rev10) with remote pneumatic actuation pressure. Note that the AFS includes trailing edge serrations, which is important to highlight as will be discussed below. Furthermore, the blade has vortex generators (VGs) mounted on the suction side. However, these VGs were removed prior to the execution of the test in question as discussed in section 4 to avoid spanwise variations of the wake velocity deficit coming from the upwash and downwash structures trailed from the VG vanes. 
[revised manuscript text omitted]

Barber, S., Deparday, J., Marykovskiy, Y., Chatzi, E., Abdallah, I., Duthé, G., Magno, M., Polonelli, T., Fischer, R., and Müller, H.: Development of a wireless, non-intrusive, MEMS-based pressure and acoustic measurement system for large-scale operating wind turbine blades, Wind Energy Science, 7, 1383–1398, https://doi.org/10.5194/wes-7-1383-2022, 2022.

435 Barlas, T., Pirrung, G. R., Ramos-García, N., González Horcas, S., Li, A., and Madsen, H. A.: Atmospheric rotating rig testing of a swept blade tip and comparison with multi-fidelity aeroelastic simulations, Wind Energy Science, 7, 1957–1973, https://doi.org/10.5194/wes-7-1957-2022, 2022.

[revised manuscript text omitted]

---

## Author Comment (AC3)

**Comments to Reviewer #3**
Dear Reviewer

First of all we would like to thank you for your constructive review of our article. Please find below our responses (in red) to your comments (in blue) and article updated text (in brown). Please find also attached a marked-up version showing all changes in the article.

Yours sincerely

H. Aa. Madsen and co-authors

Summary of major issues : The paper present an important work with wind tunnel, tests ring and field test measurements, that are however most of the time not sufficiently described, not sufficiently justified and analyzed. This is also difficult to make a link between wind tunnel tests/ test ring and wind turbine tests as there is either no quantitative results, only the Cl/Cd curve or only the Cl/Cd time series . . . I recommand authors to maybe focus more on one set of tests and to provide a more detailed analysis.

Response:
We acknowledge that the reviewer finds that the article presents important work. However, we do not understand the argument of the reviewer that the link between wind tunnel tests and turbine tests is difficult to follow due to the lack of quantitative results. Figures 9, 10, 11, 12, and 13 present all quantitative results.

The key objective of the article is to compare the airfoil characteristics measured in the wind tunnel and on the full-scale rotor presented in Figures 10, 11, 12, and 13. The test on the rotating test rig was just included and briefly described because we find that it is an important part of the development and research work that led to the application of the wake rake on a full-scale turbine.

However, to make it clear that the main objective is to compare the full-scale rotor airfoil characteristics (Cl and Cd) with wind tunnel data, we have rewritten and shortened Sections 2.2.1 and 2.2.2 to show that in the present article we just want to describe that this part is to test the functionality and operation of the wake rake in a rotating environment with turbulent inflow.

1-Introduction : Authors underline the lack of knowledge on blade aerodynamics on solely three mechanisms : high Reynolds numbers, roughness state at the surface of the blades, outer-flow state (turbulence inflow) that acts on the laminar to turbulent transition on the airfoil surface. However, differences between wind tunnel and field tests can come from more than these 3 mechanisms. Indeed, wind tunnel turbulent inflow do not necessarily match inflows encountered in blade real operating conditions and not only regarding turbulence level. Mean inflow velocity gradients combined with different turbulent inflows may lead to different operating AoA. This can induce important load variations and even flow separation (even small) which induces 3D and unsteady flow over the airfoil and thus load variations. When the flow separates other aspects such as the airfoil aspect ratio, and the centrifugal forces play an important role in differences with wind tunnel measurements. Centrifugal forces are reproduced by the rotating test ring, however, the

aspect ratio is very low (2) on both wind tunnel and rotating test ring.

Response:
First we mention that we did not write **"solely"**, but we write: *... in the lack of knowledge of, **"e.g."** the characteristics of the .......*

However, we propose to change the original formulation:

"Together, these three mechanisms have an important influence on the boundary layer transition characteristics."

Changed to:

"Together, these three mechanisms have an important influence on the boundary layer transition characteristics, but other effects like three-dimensional flow effects in combination with local separation are also important."

You write further:

Please reformulate hypothesis of this study in the introduction : - The author place themselves in the hypothesis with no inflow inhomogeneity (mean or turbulent) along the blade in real operation (there is no inflow measurements able to measure the presence of any mean or unsteady velocity gradient). Only two points are measured in the near blade inflow (in the blade induction zone?) using the 5 hole pressure sensors. - The AoA is sufficiently low for the flow to stay attached.

Response:
If we understand correctly this is the imagination of the rotor in wind tunnel flow, e.g. like the experiment NREL carried out in year 2000 placing a 10 m rotor in the giant NASA Ames wind tunnel. In this case, comparing the characteristics of the wind tunnel and the rotor airfoil characteristics would show the impact of stall, three-dimensional flow effects, and centrifugal effects among others. Back then, there was much focus on the stall effects as much of the turbines were stall-regulated.
However, in our present experiment we have taken a big step further and measured the airfoil characteristics on a pith-regulated full-scale rotor operating below stall except in few instantaneous moments. Previous detailed measurements have pointed to the impact of atmospheric turbulence moving forward the point of transition and, e.g., contributing to the increase in drag.
We therefore find that the present introduction describes the key objective of the article in a clear way.

2 - Experimental set-up :
Figure 1 is unclear. It is very difficult to see how the rake is installed relatively to the blade. Please provide a clearer view (maybe add a sketch).

Response:
We agree that it might be difficult to see the wake rake setup in the wind tunnel. This is mainly

because the wind tunnel was in an acoustic version with fiber cloth semi-transparent walls.

We propose to remove the photos.

Due to all the instrumentation (wake rake plus 5 hole pressure sensors), the flow over the airfoil might be 3D even if not separated (low AoA). This could happen especially due to the low aspect ratio (equal to 2).
How the two dimensionality is checked (wood tuft ? Visualization??) ?

Response:
The flow over the blade section is certainly three-dimensional over most of the span but at the mid span section where the pressure taps are installed the flow is expected to be approximately two-dimensional.

However, as mentioned above, Section 2.2.2 has been shortened and does not longer included measurement of airfoil characteristics.

The wake rake composition is not given clearly in section 2.2. A picture appears only in L120, with no explanation on the location of « static tubes », « total pressure tubes » or « head of pitot tubes » that seems to compose the wake rake as given from L116 to L119. Please provide a clear description/composition of the wake rake (on figure 2), at section 2.2.
Please justify the choice of this composition.

Response:
We have inserted text on the photo of Figure 2 marking the four static probes and the two different spacings of 5 and 10 mm between the total probes.

The geometry of the airfoil section is not given. This is a very important matter as the state of the flow over the airfoil depends largely on the airfoil geometry. $15\deg$ is certainly a separated state, and depending on the airfoil geometry (thickness, curvature), this correspond to a deep stall or a slight trailing edge flow separation ... Also, there is no means to evaluate the drag without the wake rake (no reference case). This could be a an xfoil simulation (valid only at low AoA) or drag measurements using a balance system.
Please provide the airfoil geometry that is tested, and at least a reference case with measurement of Cl, Cd versus AoA.

Response:
The airfoil is a 21% thick airfoil designed specifically for wind turbine application by Siemens Gamesa and the design is confidential. However, Section 2.2.2 has been rewritten and shortened and does not longer include measurement of airfoil characteristics so that the details of the airfoil shape are not important.

L110-114 : « measurement is only slightly affected by ». How can you conclude on this as there is no « reference case » ? Also, please provide a more quantitative description.
• L113 « The cd is only slightly affected by the pressure orifices, but largely affected by the

five-hole Pitot tube connection »
This is to me a major issue for drag measurement. Please develop more for which configurations this sentence applies ? Please provide a quantification of this impact. At last, justify more why using 5 hole pressure sensors, and give possible solutions for future work. • Nothing is said on impact of AoA (8° and 15° were tested) on drag measurements

Response:
As mentioned in the description of the test parameters, the wake rake was tested in the following different vertical heights:

– Vertical position: 1.0 m (behind pressure orifices), 1.15 m (slightly above pressure orifices), 1.38 m (maximum height, closer to the connecting tubes for the five-hole Pitot tube).

It is on the basis of these measurements that the conclusions were made.

As concerns the drag from the five hole pitot tube we do not consider this as a problem as we position the five hole pitot tube well away from the measurements of the drag, e.g. assuming the wedge of the wake from the five hole pitot tube has angle of +-10 deg.

Nothing is said on impact of AoA (8° and 15° were tested) on drag measurements Please provide Cd error quantification for both AoA.

Response:
Not relevant any more as the graph with CD characteristics from the RTR has been deleted.

Section 2.2.2 • L136 « Detailed tests were carried out in different wind conditions ... » Please provide detailed informations on tested configurations (give values of velocity, pitch and flap angles, together with calculated AoA – and provide method used to compute AoA). • P6 L137 : There is no description of the VG and of the surface roughness (shape, location on the blade, wind tunnel tests etc . . . ) used for figure 4. Also this figure is out of the main paper subject. Either the authors give more information on these tests, or the figure should be removed. Also, why not providing a comparison between wind tunnel drag measurements and test ring measurements (and then field tests measurements) ? Isn't is the purpose of the paper ?

Response:
Not relevant any more as Section 2.2.2 has been rewritten with focus only on the test of the functionality of the wake rake attached to a rotating blade section.

Section 2.2.3 • L144 «
it was decided to design and manufacture a new frame for attachement of the wake rake on the full scale blade » I do understand this difficulty, however, do you have reproduced this set-up in wind tunnel (at least once) to evaluate this impact of this new mounting system ? If not, how do you expect this mounting to change results ? Please give more details.

[Figure]

Response:
We did not test the frame for the attachment of the wake rake on the full scale turbine in the PLC wind tunnel as this is physically not possible due to the size. This had at least to be done on a downscaled model.

However, we do not expect major disturbances from the frame at the three measurement positions. 1) the inflow measured with the five hole pitot tube attached at the leading edge of the blade. 2) The wake rake was positioned spanwise away from the disturbed wake flow behind the flyboard carrying the five hole pitot tube. 3) The pressure belt, although placed below the carbon circular tubes, but in a distance of several tube diameters. Also, bearing in mind that the blade was operated in attached flow in an AoA range from 0 to 10 deg. The attachment frame for the wake rake was designed to carry the centrifugal loading on the rame and the wake-rake itself, to be light-weight and non-intrusive to the blade strucutre, and to provide a highly stiff structure to avoid vibrations of the frame during the measurements. Furthermore, the space between the structural elements of the frame and the measurement instruments was kept as large as feasible.

Minor comments :
• L21-22 : « This is due to . . . 2D wind tunnel flow and unsteady, turbulent 3D flow experienced on the wind turbine blade » I agree that there exist major differences from wind tunnel experiments and field measurements, but I won't say that the inflow is 2D in wind tunnels. In wind tunnels the flow is homogeneous and with low turbulence intensity, unless customized differently (passive or active grids, gust or other generated perturbations . . . ). please be more specific

Response:
We find that this is an accepted terminology to describe wind tunnel flow as two-dimensional. Of course, three-dimensional flow exists at the interface between the blade section and the tunnel walls and at high AoA with separated flow. However, to modify the statement we propose to write:

.... different operating conditions for a 2D airfoil section tested in the steady wind tunnel flow ....

L54 : « This is commonly measured ... »
This is not clear what « This » refers to. One could think that the wake rake measure only the viscous drag. Please reformulate.

Response:
Text changed to:

The total drag (pressure drag + viscous drag) is commonly measured with a so-called wake rake measuring the momentum loss in the near wake behind the trailing edge of blade section.

• « Experimental set-up » section :
The description of experimental set-up is made of bullet points, which is unusual and rather adapted to technical reports. Please provide sentences rather than bullet points.

Response:

[Figure]

Section 2.2.1 has been rewritten and two tables inserted with the information previously presented in bullet points.

**Compare Results**

Old File:

**WES_paper_2025_75_1st_version.pdf**

24 pages (9.71 MB)

28-04-2025 07:33:32

versus

New File:

**WES_paper_2025_75_reviewed.pdf**

23 pages (7.38 MB)

14-07-2025 11:24:43

**Total Changes**

**296**

Text only comparison

**Content**

118   Replacements

100   Insertions

78   Deletions

**Styling and Annotations**

0   Styling

0   Annotations

Go to First Change (page 1)

[revised manuscript text omitted]

**2.2.1 Prototype design and wind tunnel tunnel testing**

Wind tunnel tests were conducted in the PLCT to estimate the main dimensions of the wake rake design. However, it should be noted that the design was also based on the already developed wake rake installed as part of the standard measurement system in the PLC wind tunnel at DTU.

The test section in the PLCT is 2 m in span and 3 m width. For this campaign, a blade section with a 1 m chord and a 2 m span was used. The blade section was equipped with a trailing edge morphing flap, as one of the crucial design parameters

| Wake rake wind tunnel tests | |
|---|---|
| Tunnel flow speed | 20 m/s (Re $= 1.3 \times 10^6$) and 30 m/s (Re $= 1.9 \times 10^6$) |
| Distance behind TE | 0.1 m, 0.3 m, 0.5 m and 1.0 m |
| Vertical position | 1.0 m, 1.15 m and 1.38 m |
| Angle of attack | 8 and 15 deg |
| Flap deflection | 0 mm, -20 mm and 20 mm |

**Table 1.** Overview of the parameters varied during the wind tunnel tests forming the basis for the final wake rake design. The three vertical positions were chosen to be: 1) behind pressure orifices (1.0 m); 2) slightly above pressure orifices (1.15 m) and 3) maximum height, closer to the connecting tubes for the five-hole Pitot tube (1.38 m).

to determine in the PLCT tests is the width of the wake rake so it can capture the deficit with the flap in its maximum and minimum deflection positions. The instrumentation comprised standard pressure orifices in the midspan and two five-hole Pitot tubes at 0.5-m and 1.5-m span positions measured from the floor of the tunnel.

The parameters varied during the wind tunnel tests are listed in Table 1. The test parameters in the wind tunnel were chosen to match conditions similar to the wake rake test on the rotating test rig.

The drag coefficient $c_d$ is found using the method of Jones (1936) and may be expressed as:

$$c_d = 2 \int\limits_{y_{min}}^{y_{max}} \sqrt{c_{p,tot} - c_{p,stat}} \left[1 - \sqrt{c_{p,tot}}\right] dy, \tag{1}$$

with

$$c_{p,tot} = \frac{H_0 - p_0}{0.5\rho U_0^2}, c_{p,stat} = \frac{p - p_0}{0.5\rho U_0^2}, \tag{2}$$

where $y_{\max} - y_{\min}$ is the length of the integration path of the wake rake velocity deficit, $p_0$ the static pressure, $p$ the dynamic pressure, $H_0$ the total head, $\rho$ the density, and $U_0$ the velocity of the free stream.

However, we use the same formula for the derivation of the drag coefficient in turbulent flow, and the accuracy of this needs to be further investigated in future tests.

From the wind tunnel test results, the following conclusions and parameter settings were derived: 1) The $c_d$ measurement is only slightly affected by the downstream distance from the trailing edge (0.1 m to 0.5 m); 2) The $c_d$ measurement is only slightly affected by rotating the rake with the profile, at least not for the value of angle of attack of 8 deg (It is also a matter of making the end of the tube conical); 3) The $c_d$ is only slightly affected by the pressure orifices, but largely affected by the five-hole Pitot tube connection and 4) The wake width is approximately 0.3 m at $\alpha = 8$ deg for a flap deflection of $\delta = \pm 20$ mm..

| | Final wake rake dimensions |
|---|---|
| Position of the head of Pitot tubes | 0.3 m behind the TE |
| Wake rake width | 0.4 m (0.1 m wider than the width measured in tests) |
| Static tubes | 4 in line with the total pressure tubes |
| Total head tubes | 56 with 5 mm and 10 mm spacing, respectively |

**Table 2.** Overview of the final wake rake design parameters. See also the photo of the wake rake in Fig.1.

Based on the findings and conclusions of the wind tunnel tests, the final design parameters as shown in Fig. 2 were chosen. A photo of the manufactured wake rake is shown in Fig. 1. Inside the wake rake plastic tubes are connected to the pressure probes and led out at both ends of the rake to the pressure scanners outside.

[Figure]

**Figure 1.** Photo of the final manufactured wake rake. See also the parameters and dimensions in Table 2.

**2.2.2   Mid scale testing of the functionality and integration of the wake rake on the blade section on a rotating test rig**

To gain experience with the functionality and operation of the wake rake in a rotating environment before going to the full scale turbine application, the wake rake was installed and tested on a rotating test rig (RTR) Madsen et al. (2015), Ai et al. (2019) and Barlas et al. (2022). The RTR is built on a 100 kW turbine platform where the rotor is replaced by a 10 m pitch-able long boom carrying a 2 m span blade section at the tip, see the right photo in Fig. 2. In fact, a blade section for tests in the PLC tunnel can be tested directly on the RTR.

The wake rake was mounted on the blade section with a frame of circular carbon tubes attached to the aluminum frame inside the blade section; see the photos in Fig. 2. The 2 m long rectangular blade section + end caps has an active trailing edge flap system with serrations like in the full scale experiment. However, the serrations were covered with tape as it is expected that the flow will not be uniform in the position of the wake rake due to the serrations and thus make the drag measurement uncertain. The wake rake probes were connected to four 16 channel differential pressure scanners positioned inside the blade section and with the reference port connected to the static port of one of the five hole pitot tubes attached the leading edge of the blade section.

The RTR tests confirmed that the wake deficit of the blade section could be satisfactorily covered within the wake rake width even in the turbulent inflow and with flap activation. The tests also confirmed that the carrying frame was stable. .

[revised manuscript text omitted]

In what follows a short description of the AFS is given, but the reader is referred to (Gomez Gonzalez et al., 2022, 2023) for further details. The outer portion of the blade (span location 38 m – 58 m) is retrofitted with the active flap (internal reference FT008rev10) with remote pneumatic actuation pressure. Note that the AFS includes trailing edge serrations, which is important to highlight as will be discussed below. Furthermore, the blade has vortex generators (VGs) mounted on the suction side. However, these VGs were removed prior to the execution of the test in question as discussed in section 4 to avoid spanwise variations of the wake velocity deficit coming from the upwash and downwash structures trailed from the VG vanes. 
[revised manuscript text omitted]

Barber, S., Deparday, J., Marykovskiy, Y., Chatzi, E., Abdallah, I., Duthé, G., Magno, M., Polonelli, T., Fischer, R., and Müller, H.: Development of a wireless, non-intrusive, MEMS-based pressure and acoustic measurement system for large-scale operating wind turbine blades, Wind Energy Science, 7, 1383–1398, https://doi.org/10.5194/wes-7-1383-2022, 2022.

435 Barlas, T., Pirrung, G. R., Ramos-García, N., González Horcas, S., Li, A., and Madsen, H. A.: Atmospheric rotating rig testing of a swept blade tip and comparison with multi-fidelity aeroelastic simulations, Wind Energy Science, 7, 1957–1973, https://doi.org/10.5194/wes-7-1957-2022, 2022.

[revised manuscript text omitted]

---

## Author Response (AR3)

**Reviewer 3**

The reviewer would like to thank the authors for considering the suggestions.

In reply to the response please consider the following:

-In the derivation of Jones' method, which considers 3 sections (upstream (0), at wake rake (1) and far downstream (2)), it is assumed that the static pressure at sections (0) and (2) are equal, which can be challenged in a rotor application. Now the authors mention that Jones' method still holds because the static pressure is being measured also in the rotor plane (and not at an undisturbed upstream position). However the formula displayed in the manuscript still resembles Jones' original equation and it is not clear where the static pressure from the pitot comes in or how this validates the usage of the original equation. Therefore, it is anticipated that static pressure gradients due to rotor blockage can affect the determination of drag coefficient in this setting.

Reply: see extended text (red) in 2.2.1.

**-Unsteady characteristics**

It is recommended to clarify the reduced frequency in the manuscript to get a sense for the unsteadiness.

Reduced frequency based on 1p = 0.013 which means we are in the quasi-steady regime. Reply: Inserted in text (red) in 2.3.6.

**-Drift**

It would be worthwhile to add the considerations regarding drift in the manuscript.

Reply: - from data sheet, drift <1mbar / year Inserted in modified text (in red) in 2.3.4-

**-RTR**

Pity that the RTR results are removed, as it could shed more light on the observed drag offset. But I suppose there is more than enough material for publication with wind tunnel and field test already.

Reply: None.

**-Parasitic drag**

The authors mention that the effects of the instrumentation on the power and load levels of the turbine are minimal. It would be good to substantiate this claim based on arguments or measurements.

**Reply:**

If we assume a small part of the blade span at the instrumentation, e.g. 0.25 m has an increased drag coefficient of e.g. 2 due disturbances from instrumentation, it gives of power loss of 1.13 kW which is minimal for a 4 MW rotor.

**Blade surface pressure and drag measurement of a blade section on a 4.3 MW turbine with trailing edge flaps**

Helge Aagaard Madsen1, Alejandro Gomez Gonzalez2, Thanasis Barlas1, Anders Smærup Olsen1, Sigurd Brabæk Ildvedsen1, and Andreas Fischer1

Correspondence: Helge Aagaard Madsen (hama@dtu.dk)

Abstract. In this paper we present the measurements of local aerodynamic sectional characteristics on a full-scale rotor blade with a novel add-on instrumentation comprising a wake rake, a pressure belt, and a five hole Pitot tube. The general objective of the research work is to provide information on the differences between airfoil performance in wind tunnel flow and on a full-scale rotor. Although pressure belt measurements have been performed in earlier studies, this is the first campaign to use a wake rake at the full-scale level. We present the wake rake development and testing in the wind tunnel and on a rotating test rig which finally led up to installation on the 4.3 MW turbine. A more specific objective with the campaign has been to characterize the aerodynamic performance of a trailing edge active flap system installed on one of the blades. The short measurement campaign of two days comprised measurements of the flaps actuated at constant time intervals of 60 s between fully retracted and activated with a control set point of 75% of full deflection. The lift and drag characteristics are compared with a similar flap actuation in a wind tunnel experiment. Both the relative change in lift and drag coefficients as a function of flap actuation correlate well with wind tunnel measurements, but the absolute drag levels measured on the rotor are higher than the wind tunnel data. During the measurement campaign, it was also demonstrated that it is possible to clearly measure the increased drag from adding a roughness tape at the leading edge of the airfoil. This illustrates the potential use of the measurement system to capture the effect of variations of the local aerodynamic performance at full-scale even for elements at boundary layer scale, e.g. the impact of roughness or the positioning of add-ons such as vortex generators.

**1 Introduction**

The lack of detailed measurements to characterize the aerodynamics and aeroelasticity of full-scale turbines is a barrier to further reliable turbine development and upscaling (Veers et al., 2023). In particular, inflow and blade surface pressure measurements on turbines operating in turbulent atmospheric inflow are crucial data for a deeper understanding and insight into how the airfoil characteristics (lift, drag, and moment coefficients) measured in wind tunnels can be transferred and used for the design of full-scale wind turbine blades. This is due to the quite different operating conditions for a 2D airfoil section tested in the steady wind tunnel flow and the unsteady, turbulent 3D flow experienced on the wind turbine blade.

<sup>1Department of Wind and Energy Systems, Technical University of Denmark, 4000 Roskilde, Denmark

<sup>2Technology Development, Siemens Gamesa Renewable Energy, 7330 Brande, Denmark.

[revised manuscript text omitted]

**2.2.1 Prototype design and wind tunnel tunnel testing**

Wind tunnel tests were conducted in the PLCT to estimate the main dimensions of the wake rake design. However, it should be noted that the design was also based on the already developed wake rake installed as part of the standard measurement system in the PLC wind tunnel at DTU.

The test section in the PLCT is 2 m in span and 3 m width. For this campaign, a blade section with a 1 m chord and a 2 m span was used. The blade section was equipped with a trailing edge morphing flap, as one of the crucial design parameters

|                    | Wake rake wind tunnel tests                                           |
|--------------------|-----------------------------------------------------------------------|
| Tunnel flow speed  | 20 m/s (Re $= 1.3 \times 10^6$ ) and 30 m/s (Re $= 1.9 \times 10^6$ ) |
| Distance behind TE | 0.1 m, 0.3 m, 0.5 m and 1.0 m                                         |
| Vertical position  | 1.0 m, 1.15 m and 1.38 m                                              |
| Angle of attack    | 8 and 15 deg                                                          |
| Flap deflection    | 0 mm, -20 mm and 20 mm                                                |

**Table 1.** Overview of the parameters varied during the wind tunnel tests forming the basis for the final wake rake design. The three vertical positions were chosen to be: 1) behind pressure orifices (1.0 m); 2) slightly above pressure orifices (1.15 m) and 3) maximum height, closer to the connecting tubes for the five-hole Pitot tube (1.38 m).

to determine in the PLCT tests is the width of the wake rake so it can capture the deficit with the flap in its maximum and minimum deflection positions. The instrumentation comprised standard pressure orifices in the midspan and two five-hole Pitot tubes at 0.5-m and 1.5-m span positions measured from the floor of the tunnel.

The parameters varied during the wind tunnel tests are listed in Table 1. The test parameters in the wind tunnel were chosen to match conditions similar to the wake rake test on the rotating test rig.

The drag coefficient  $c_d$  is found using the method of Jones (1936) and may be expressed as:

$$c_d = 2 \int_{y_{min}}^{y_{max}} \sqrt{c_{p,tot} - c_{p,stat}} \left[ 1 - \sqrt{c_{p,tot}} \right] dy, \tag{1}$$

95 with

105

$$c_{p,tot} = \frac{H_0 - p_0}{q_0}, c_{p,stat} = \frac{p - p_0}{q_0},$$
(2)

where  $y_{\rm max}-y_{\rm min}$  is the length of the integration path of the wake rake velocity deficit,  $p_0$  the static pressure, p the dynamic pressure,  $H_0$  the total head,  $\rho$  the density, and  $q_0$  the free stream dynamic pressure. The total and static pressures are measured by a differential pressure scanner so the measurement gives the  $H_0-p_0$  and  $p-p_0$  directly. The free stream dynamic pressure,  $q_0$ , is estimated from the two outer static and total probes, as the  $q_0$  based on the five hole Pitot tube proved to be questionable.

The Jones formula is derived for uniform, steady, and two-dimensional flow. The use of the same formula in the present case behind a blade section of a rotating full-scale blade in atmospheric flow thus deviates from the flow conditions for which the formula was derived. However, it is still assumed that the Jones formula is applicable in the rotating reference frame of the turbine, as the flow curvature at the wake rake location is very small compared to the blade section chord (chord/radial position of wake rake  $\approx 1/50$ ). Therefore, the curvature effects can be neglected and a two-dimensional flow can be assumed at the wake rake location. The impact and uncertainty of applying the Jones formula in the turbulent-flow field still need to be further

|                                     | Final wake rake dimensions                           |
|-------------------------------------|------------------------------------------------------|
| Position of the head of Pitot tubes | 0.3 m behind the TE                                  |
| Wake rake width                     | 0.4 m (0.1 m wider than the width measured in tests) |
| Static tubes                        | 4 in line with the total pressure tubes              |
| Total head tubes                    | 56 with 5 mm and 10 mm spacing, respectively         |

**Table 2.** Overview of the final wake rake design parameters. See also the photo of the wake rake in Fig.1.

investigated in future research, e.g. by extracting detailed pressure and flow details around the blade section from full resolved CFD simulations of the rotor. In short, extract the sectional drag coefficient from a numerical simulation and compare it with the drag coefficient for the section in two-dimensional flow.

From the wind tunnel test results, the following conclusions and parameter settings were derived: 1) The  $c_d$  measurement is only slightly affected by the downstream distance from the trailing edge (0.1 m to 0.5 m); 2) The  $c_d$  measurement is only slightly affected by rotating the rake with the profile, at least not for the value of angle of attack of 8 deg (It is also a matter of making the end of the tube conical); 3) The  $c_d$  is only slightly affected by the pressure orifices, but largely affected by the five-hole Pitot tube connection and 4) The wake width is approximately 0.3 m at  $\alpha$  = 8 deg for a flap deflection of  $\delta$  =  $\pm$  20 mm..

Based on the findings and conclusions of the wind tunnel tests, the final design parameters as shown in Fig. 2 were chosen. A photo of the manufactured wake rake is shown in Fig. 1. Inside the wake rake plastic tubes are connected to the pressure probes and led out at both ends of the rake to the pressure scanners outside.

**2.2.2 Mid scale testing of the functionality and integration of the wake rake on the blade section on a rotating test rig**

To gain experience with the functionality and operation of the wake rake in a rotating environment before going to the full scale turbine application, the wake rake was installed and tested on a rotating test rig (RTR) Madsen et al. (2015), Ai et al. (2019) and Barlas et al. (2022). The RTR is built on a 100 kW turbine platform where the rotor is replaced by a 10 m pitch-able long boom carrying a 2 m span blade section at the tip, see the right photo in Fig. 2. In fact, a blade section for tests in the PLC tunnel can be tested directly on the RTR.

125

130

The wake rake was mounted on the blade section with a frame of circular carbon tubes attached to the aluminum frame inside the blade section; see the photos in Fig. 2. The 2 m long rectangular blade section + end caps has an active trailing edge flap system with serrations like in the full scale experiment. However, the serrations were covered with tape as it is expected that the flow will not be uniform in the position of the wake rake due to the serrations and thus make the drag measurement uncertain. The wake rake probes were connected to four 16 channel differential pressure scanners positioned inside the blade section and with the reference port connected to the static port of one of the five hole pitot tubes attached to the leading edge of the blade section.

**Figure 1.** Photo of the final manufactured wake rake. See also the parameters and dimensions in Table 2.

The RTR tests confirmed that the wake deficit of the blade section could be satisfactorily covered within the wake rake width even in the turbulent inflow and with flap activation. The tests also confirmed that the carrying frame was stable. See the video of the test of the wake rake on the rotating test rig, https://doi.org/10.5281/zenodo.17062097.

**135 2.2.3 Wake rake attachment on the full scale blade**

[revised manuscript text omitted]

In what follows a short description of the AFS is given, but the reader is referred to (Gomez Gonzalez et al., 2022, 2023) for further details. The outer portion of the blade (span location 38 m - 58 m) is retrofitted with the active flap (internal reference FT008rev10) with remote pneumatic actuation pressure. Note that the AFS includes trailing edge serrations, which is important to highlight as will be discussed below. Furthermore, the blade has vortex generators (VGs) mounted on the suction side. However, these VGs were removed prior to the execution of the test in question as discussed in section 4 to avoid

(b) Close-up view of the wake-rake setup.

(c) Close-up view of serrations covered with a tape to avoid spanwise velocity deficit variations.

[revised manuscript text omitted]